

# An advanced error state Kalman filter (ESKF)-based terrain contour matching (TERCOM) method for tracking an aerial vehicle using a low-cost digital elevation map

Muhammad Bilal Kadri[1] and Sofia Yousuf[2]

[1] College of Computer & Information Science (CCIS), Prince Sultan University, Riyadh, Saudi Arabia
[2] College of Engineering, Karachi Institute of Information Technology, Karachi, Sindh, Pakistan

## ABSTRACT

Terrain Aided Navigation (TAN) systems hold significant potential for delivering accurate navigation for Uncrewed Aerial Vehicles (UAVs). However, a major limitation of conventional TAN systems lies in the time-consuming correlation technique used to search the *a priori* map, specifically the Digital Elevation Maps (DEM). This article presents a fuzzy heuristic method for the mean absolute deviation (MAD) correlation scheme (FH-MAD), aimed at reducing the computational complexity and execution time of the TAN algorithm. The fuzzy logic system uses heading and roll angle data from onboard sensors to determine the aircraft's matching area. The output membership functions are designed based on parameters that depend on terrain features. Additionally, the proposed method incorporates an error state Kalman Filter (ESKF) as the navigation algorithm to estimate the UAV's position under various maneuvering conditions. To evaluate the effectiveness of the proposed system, tests were conducted using two distinct DEMs with varying topographical characteristics and dimensions. The results demonstrate improved position accuracy and a significant reduction in computation time compared to traditional TAN methods, making the approach suitable for real-time UAV navigation applications.

# INTRODUCTION

Uncrewed Aerial Vehicles (UAVs) have become one of the most important and promising area for both civilian as well as military-based applications. In this perspective, the localization and tracking information of a UAV are vital to provide the UAV with a complete navigation solution (*Yigit & Yilmaz, 2022*; *Ali et al., 2021*; *Abdelkader et al., 2025*). The main purpose of a navigation system is to determine the geographical position, altitude, orientation and velocity of an aircraft in order to accomplish its flight from one point to another and good control performance (*Bousbaine et al., 2023*). The Inertial

Corresponding author
Muhammad Bilal Kadri, mkadri@psu.edu.sa

Navigation System (INS) is one of the most popular technologies for providing direct aircraft navigation. This system is employed to measure navigation parameters including the three dimensional position, velocity and acceleration (PVA) information. The main advantage of the system is that it is robust towards external influences. The drawback of the system is that the outputs are computed using integration methods resulting in the path integral errors which accumulate over time (*Han et al., 2018*). Another navigation system is the Global Positioning System (GPS), which is a satellite-based navigation technology. The GPS was designed to overcome the deficiencies of other positioning systems. These systems have been used in conjunction with the INS and the Kalman Filter (*Liouane et al., 2022*) technology to correct the INS positioning errors (*Yousuf & Kadri, 2020*). The GPS is prone to signal loss, jamming issues and multi-path errors (*Carroll & Canciani, 2021*). Therefore, in order to cope with the disadvantages of both the sensors, the systems based on Terrain Reference Navigation (TRN) have become an attractive alternative solution for aircraft positioning and localization which is the main motivation for the study conducted in this work.

TRN systems are passive systems which are independent of GPS and therefore are resistant to the external factors (*Lee & Bang, 2018a*), hence these systems are attractive towards the requirements for military-based localization applications. Furthermore, the most important feature of TRNs is that it can operate in all weather conditions, day and night as well as lower and higher altitudes (*Vaman, 2012*). The performance and accuracy of these systems is mainly limited by the terrain features for instance, the terrain roughness and uniqueness characteristics. More recently, the concept of fuzzy logic (FL) has been applied to increase the positioning accuracy of a TRN system (*Shaukat, Moinuddin & Otero, 2021*; *Lin, Yu & Wu, 2021*). The FL theory is effective in providing flexibilities for reasoning, considering the factors such as uncertainty, inaccuracy, subjectivity and imprecision. Also, it has an excellent ability to deal with unreliable or incomplete knowledge bases (*Gallab et al., 2019*). In this article, the problem of positioning accuracy and computational complexity of the Terrain Contour Matching (TERCOM) algorithm for a UAV is studied, utilizing a fuzzy-based approach. In this method, a fuzzy-based heuristic mean absolute deviation (MAD) algorithm is designed for TERCOM strategy. For aircraft navigation, the method based on error state Kalman Filtering (ESKF) technology is employed. Since, in most of the cases the error profile of the non-linear system (such as a UAV), the system states have complex behavior, hence error in the states can be predicted using ESKF. In studies, it has been reported that the ESKF is more robust than the conventional EKF filter (*Xu et al., 2022*; *Lee & Bang, 2018b*). In the literature, several advantages of ESKF have been reported compared to the EKF (*Shaukat, Moinuddin & Otero, 2021*; *Wang et al., 2020*): (1) Singularity issues: in an ESKF, the singularity issues are avoided in the covariance matrices which result due to redundancy and over parameterization. (2) validity of linearization: the ESKF is always operated close to origin, therefore, the issues related to gimbal lock issues, parameter singularities are avoided. This guarantees that the validity of the linearization assumption holds at all times. (3) Faster computation of jacobians: in an ESKF, the computation of the Jacobians is fast since the error states are very small and hence all the second order products are negligible.

(4) Observability of errors: the ESKF correction rates are slower than the predictions rate. This is due to the fact that the error dynamics itself are slower and the large scale dynamics are incorporated in the nominal state. This means that the corrections are made less frequently as compared to the EKF-based estimation method. (5) Retuning of covariance matrices: the greatest advantage of the ESKF model is that it does not requires the re-tuning of the process, measurements and the error noise covariance irrespective of the different suite of aircraft maneuvers. This is in contrast to the EKF model which requires retuning for the type of the flight maneuver an aircraft performs.

## State of the art review

The terrain contour matching (TERCOM) is a class of TRN algorithm which uses pre-stored terrain elevation data to determine the current location of the aircraft. The history of the TERCOM system dates back fundamentally to the design of a cruise missile system in the late 1950s. The cruise missile system is a pilotless, continuously powered and a dispensable vehicle, which is specifically used for delivering the nuclear devices. One of the major advantages of the TERCOM technique is that it can achieve faster localization with accuracy in cases of large initial positioning error (*Peng et al., 2018*). The disadvantage is that the technique has poor real-time performance. Another TRN algorithm is the Sandia Inertial Terrain Aided Navigation (SITAN), which uses the Extended Kalman Filtering (EKF) method (*Jayaramu et al., 2021*). SITAN provides the capability of real-time positioning, however, the major disadvantage is that it produces high false-fixes in cases of large initial positioning errors (*Dai & Kang, 2014*).

In the recent years, hybrid TAN strategies have also been reported combining TERCOM and Sandia Inertial Terrain Aided Navigation (SITAN) algorithms to exploit the advantages of both the technologies in order to meet the military requirements. The Terrain Profile Matching (TERPROM) is a hybrid two-phase TRN method which uses TERCOM strategy in the acquisition mode and SITAN algorithm in the tracking mode (*Eroglu & Yilmaz, 2014*). Apart from position estimation, the system offers a number of features including Predictive Ground Collision Avoidance (PGCA), Obstruction Warnings (OW), Terrain Following (TF), and Passive Ranging (PR) (*Lee & Bang, 2018a*). The disadvantage of the scheme is that it takes longer computational time due to batch processing in the TERCOM method.

The accuracy and reliability are one of the key factors characterizing a navigation system. In *Yoo et al. (2012)*, an improved TERCOM algorithm using the velocity correction method was proposed. The observability analysis was performed and a selective error method was employed to correct the velocity errors in the INS in addition to the position errors. In *Zhao (2012)*, a TERCOM strategy based on three stage logic was proposed in order to mitigate the false-fix issues in the TERCOM algorithm. In *Wu, Fei & Li (2012)*, a multi-path Terrain Contour Matching algorithm using the laser scanning altimetry replacing the traditional radar altimeter sensor. The results of the scheme verified that 2D altimetry can effectively meet the localization requirement of the UAV. The disadvantage is that the scheme is applicable for low altitude UAV flight missions only. In *Yan et al. (2018)*, a TRN system based on particle filter (PF) and adaptive scaling method for Digital

Elevation Maps (DEM) for improving the navigation precision of the UAV was proposed. The system is capable of providing good navigation accuracy in cases of high UAV altitudes. The disadvantage is that the performance of the system degrades in case of flat and repetitive terrain. Two major issues with the PF-based approach are particle degradation and particle impoverishment (*Yousuf & Kadri, 2024*). In order to mitigate this issue, a TRN method based on intelligent particle filter (IPF) was proposed in *Chai, Li & Qiao (2022)*. In this method, genetic algorithm (GA) was incorporated in the PF resampling stage to improve the robustness of the IPF. In *Veselý et al. (2022)*, a system based on point mass filter (PMF) was proposed to deal with the non-linear filtering problem arising in a TRN based navigation. The PMF-based TRN showed high estimation performance, however the major disadvantage was the computational efficiency of the method. To overcome the drawbacks of the conventional PMF, the authors in *Duník et al. (2019)* proposed a TRN system based on Rao-Blackwellised point-mass filter. The system preserves the advantages of point mass filter providing high estimation accuracy and predictable computational complexity. Similarly, in *Park & Park (2020)*, a Two Stage Point Mass Filtering (TSPF) state augmentation approach for TRN application was presented. In this method, the non-linear states, *i.e.*, the UAV latitude and longitude were estimated using the PMF whereas the linear state variable, *i.e.*, the altitude was estimated using a Linear Kalman Filter (KF). The results demonstrated that the TSPMF technique was more computationally efficient compared to the RBPMF method while the estimation performance was almost similar to RBPMF (*Veselý et al., 2022*). In *Liu, Wang & Yao (2014)*, a TRN method based on Extended Kalman Filter and B-Spline Neural Network was presented for UAV navigation. The combined strategy provided better estimation results compared to EKF based TRN. The disadvantage of the BSNN network is that it is trained by gradient-based methods which may fall into local minima during the learning process. In *Eroglu & Yilmaz (2014)*, a TERCOM method utilizing the Long Short Term Memory (LSTM) deep learning scheme was introduced. The LSTM network was trained to learn all possible flight data in order to estimate the current location of the UAV. A map reader approach which uses a filtered map was employed in training the LSTM. However, the authors report issues in the presented scheme from the aspect of real-world implementations. Similarly, in *Lee & Bang (2018b)*, a TRN scheme combining Rao-Blackwellized Particle Filter (RBPF) and LSTM network was presented to minimize the navigation accuracy of a UAV. The scheme provided precise navigation compared to the conventional RBPF based TRN approach. The disadvantage was that study related to the real-time implementation of the method was not presented. More recently, in order to improve the position accuracy of a TRN system, the concept of fuzzy logic has been proposed. In *Liu, Zhang & Huang (2021)*, a TRN was proposed in which fuzzy logic was applied to estimate the distribution of variance of particles in a Particle Filter to improve the navigation accuracy of an Autonomous Underwater Vehicle (AUV). It was reported that the proposed method in this work achieved good tracking accuracy. In *Zhao, Xu & Weiqiang (2021)*, a multi-index grey fuzzy decision making method was proposed for terrain navigability analysis for an AUV. In this method, a point

mass filter (PMF) was used to simulate the navigability of the matching areas having different terrain characteristics. The results of this method indicate feasibility of the proposed fuzzy evaluation scheme in selecting the underwater matching areas for better AUV navigation.

Furthermore, A robust Kalman filter (RKF) that handles measurement outliers through kernel density estimation (KDE) is presented in this article in *Gao et al. (2025)*. A new RKF is created by using Bayesian estimation and a logarithmic Gaussian kernel to model sudden noise changes. Experiments and simulations demonstrate how well it works to increase vehicle navigation accuracy in outlier scenarios. A set-membership hybrid Kalman filter (SM-HKF) is proposed in this article in reference (*Zhao et al., 2020*) for nonlinear state estimation under both unknown but bounded (UBB) and stochastic errors. It obtains an optimal Kalman gain that takes into consideration all uncertainties by linearizing the system and combining errors using the Minkowski sum. Simulations demonstrate that when it comes to managing both systematic and stochastic errors, SM-HKF performs better than the extended Kalman filter (EKF). In *Gao et al. (2021)*, the authors suggest a distributed optimal fusion approach based on cubature rules for UAV navigation with integrated miniature inertial measurement unit (MIMU), global navigation satellite system (GNSS) and celestial navigation system (CNS). By detecting and forecasting kinematic model errors using Mahalanobis distance, it tackles issues caused by nonlinearity. Subsystems employ modified cubature Kalman filters, and the outputs are combined to estimate the globally optimal state. Experiments and simulations verify increased navigation accuracy. In order to address measurement uncertainty in nonlinear estimation, the article in reference (*Hu et al., 2024*) suggests an indirect fuzzy robust Cubature Kalman Filter (CKF). It modifies CKF *via* a scaling matrix using a fuzzy inference system (FIS) with normalized inputs and triangular membership functions. This improves convergence and robustness. Simulations demonstrate better results than conventional fuzzy-based filters. An advanced cubature information filter for tracking multiple wideband sources indoors is presented in the article in *Zhang et al. (2019)*. To increase tracking accuracy in difficult environments, it integrates a distributed noise statistics estimator. The technique provides a reliable indoor source tracking solution and improves performance in noisy environments. A novel Gaussian filtering technique is presented in the article cited (*Gao et al., 2020*), which overcomes the drawbacks of conventional methods that depend on known system noise characteristics. Utilizing maximum, a-posterior theory-based random weighting estimations, the approach adaptively modifies weights to reduce the effects of noise. Experiments and simulations demonstrate that this strategy increases estimation accuracy over conventional techniques. To overcome the drawbacks of conventional Kalman filters, which rely on known system models and noise statistics, the authors in *Xia et al. (2020)* suggest the fitting H-infinity Kalman filter (FHKF). FHKF increases the robustness and stability of nonlinear uncertain systems by estimating coefficients *via* least weighted squares using a fitting transformation. Its exceptional performance in preserving accuracy and stability is demonstrated by simulations and real-world INS/GPS experiments.

Also, an innovative orthogonality-based robust unscented Kalman filter (IO-RUKF) for hypersonic vehicle navigation is proposed in this article in *Hu et al. (2019)*. By employing hypothesis testing to identify anomalies and a robust factor to modify the Kalman gain, it mitigates measurement errors such as outliers and non-Gaussian noise. Simulations confirm that the method increases unscented Kalman filter (UKF) robustness without compromising accuracy. Using the maximum likelihood principle, this article in *Hu et al. (2020)* suggests an adaptive UKF for vehicular INS/GPS integration that estimates process noise covariance. It improves robustness against process noise uncertainty by updating noise estimates online through the introduction of a fixed-length memory window. Experiments and simulations verify that it performs better than the typical UKF. In order to manage system nonlinearity and uncertain noise statistics, this article in *Meng et al. (2016)* suggests an adaptive UKF for direct INS/GNSS integration. It estimates and updates process and measurement noise covariances online using covariance matching. Compared to standard and adaptive-robust UKF methods, simulations and experiments demonstrate increased accuracy and robustness. In *Hu, Gao & Zhong (2015)*, a refined strong tracking unscented Kalman filter (RSTUKF) is used to propose a direct INS/GNSS integration method. Through assumption testing, it detects kinematic model errors and modifies the predicted covariance using a suboptimal fading factor. In the absence of model error, this maintains UKF accuracy while improving robustness. Experiments and simulations verify its efficacy.

## Contribution of the work

This article presents a novel integration of fuzzy logic with the error state Kalman filter (ESKF) to significantly enhance the positioning accuracy and reduce the computational cost of the TERCOM navigation algorithm. Unlike traditional approaches that rely on full DEM scans for terrain correlation, we introduce a fuzzy-based heuristic mean absolute deviation (FH-MAD) method that intelligently restricts the search space using real-time aircraft dynamics, leading to faster and more efficient localization. Specifically, the key contributions of this work are as follows:

- **Fuzzy-Enhanced Terrain Matching:** we develop a fuzzy logic-based extension of the MAD algorithm (FH-MAD), which adaptively prioritizes terrain correlation regions based on flight parameters, resulting in improved positioning decisions.
- **Region-of-Interest (ROI) Selection using Flight Attitude:** the proposed method dynamically selects a rectangular sub-region of the DEM using aircraft yaw and roll angle conditions, significantly reducing the computational burden compared to exhaustive DEM searches.
- **Improved Localization Accuracy:** the FH-MAD integrated with ESKF demonstrates superior accuracy in UAV localization, validated through rigorous performance metrics under varying flight scenarios.
- **Significant Computational Gains:** our method achieves a notable reduction in execution time when compared with the conventional MAD-based TERCOM algorithm, making it viable for real-time onboard processing.

- **Robust ESKF Design for Maneuvering UAVs:** we provide a carefully designed and tuned ESKF framework tailored to handle dynamic flight conditions, ensuring consistent performance during complex UAV maneuvers.
- **Real-Time Application Readiness:** the proposed approach is well-suited for real-time UAV navigation in GPS-denied environments, offering a practical and scalable solution for autonomous aerial systems.

The rest of the article is structures as follows: in "System Overview", a brief overview of the proposed TRN method is presented. In "Proposed Method", a detailed discussion on the proposed hybrid strategy based on fuzzy-based heuristic and ESKF is provided. The simulation tests and analysis is provided in "Simulation Results". Finally, the conclusions are given in "Discussion".

## SYSTEM OVERVIEW

The complete block diagram of the proposed method is shown in Fig. 1. The TERCOM algorithm proposed in this work utilizes the improved version of the mean absolute deviation (MAD) techniques as the main matching algorithm for comparing the terrain heights obtained by taking the difference between the radar heights measurements and the barometer. The algorithm comprises of the three major modules (1) the strap-down inertial navigation system (INS) which defines the kinematic model, (2) the error state Kalman filter (ESKF) which is used to correct the INS measurements and (3) the improved mean absolute deviation algorithm for comparing the height differences to extract the best position estimate from the available DEM. The algorithm in this work utilizes a low-cost DEM of a region of interest having a resolution of 1 arc s (*i.e.*, almost 30 m).

The system operates in three phases (1) the data processing phase (2) the acquisition phase and (3) the data correlation phase. In the data processing phase, the data from INS is utilized. In the data acquisition phase, the terrain height data is obtained from the fusion of radar altimeter and the barometer based height measurements (*Krishnamurthi et al., 2020*). The INS suffers from the long-term drift issues due to the path integral errors, therefore, the height estimates from the INS are corrected using the error estimates from the ESKF. In the correlation phase, the algorithm employs the fuzzy-based mean absolute deviation (MAD) algorithm to compare the terrain height estimates with the already stored DEM heights and provide the best match based on the minimum MAD value. Based on the best match, the corresponding position vector of the aircraft is estimated from the stored DEM. The latitude longitude and height measurements from the INS are obtained by incorporating the error state Kalman filter (ESKF) which updates the erroneous INS measurements. The UAV position is obtained by extracting the two dimensional latitude and longitude vector from the available DEM corresponding to the best MAD value based on the height difference from radar and barometric measurements.

Although FH-MAD is a technique to address uncertainties that impact the accuracy of the Kalman filter, the suggested fuzzy logic approach does not directly enhance it. On the contrary, FH-MAD contributes towards the overall TAN scheme. The motivation for adopting an error state Kalman filter (ESKF) instead of a conventional Kalman filter is that

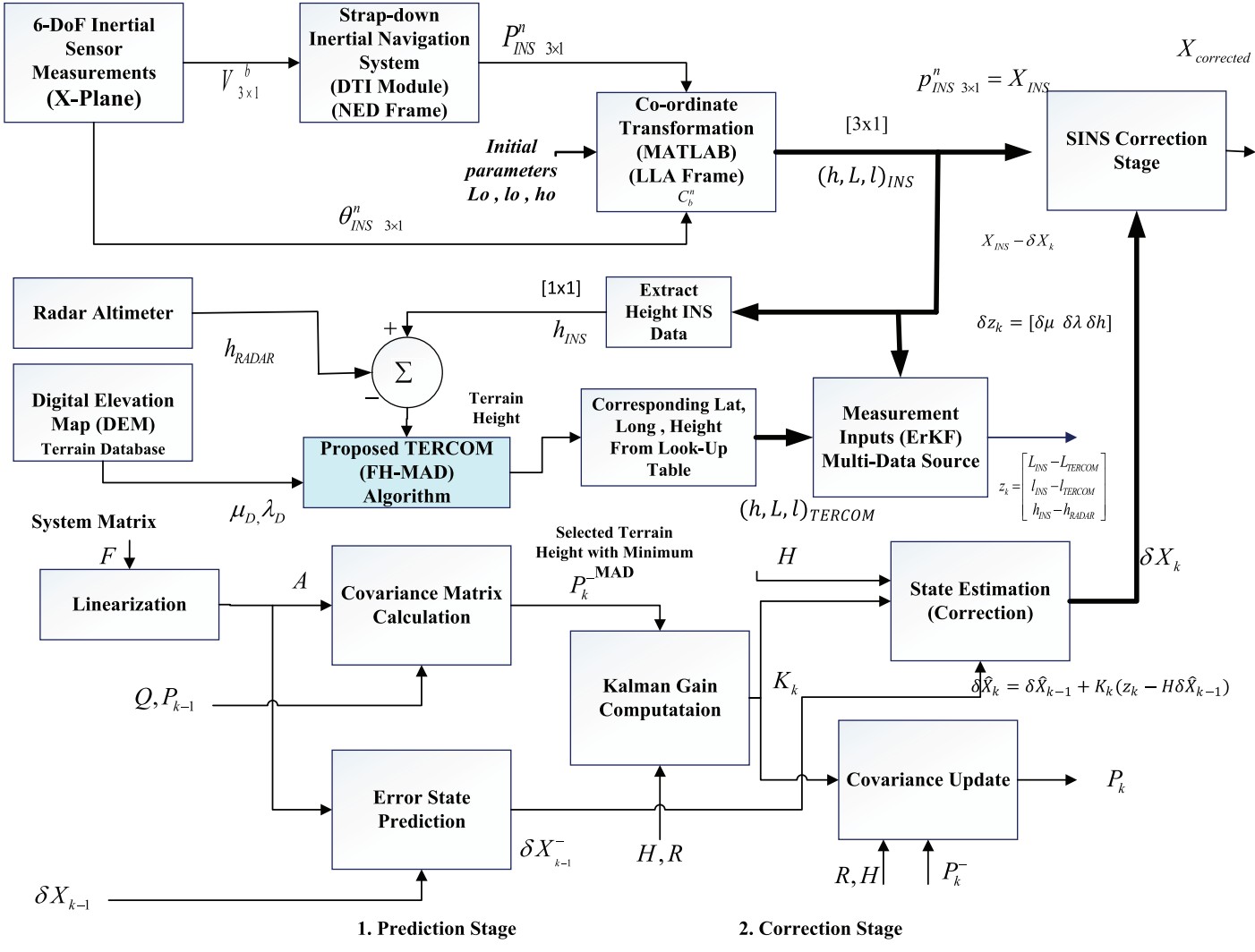

**Figure 1 Flow chart of the proposed integration scheme.**

a "simple" EKF, which linearizes the full large-magnitude state, is highly sensitive to mis-tuned noise statistics, making it prone to outliers and model-error-induced divergence.

ESKF confines linearization, tuning and consistency checks to a local error space whose variables are always small and nearly Gaussian, which keeps the linear approximation valid even under aggressive motion, makes adaptive Q/R tuning straightforward, provides numerically stable covariances for long missions, and enables reliable innovation-based outlier rejection. Consequently, when the TAN system faces real-world uncertainties *e.g.*, DEM artefacts, IMU bias drift, wind-induced modelling errors, *etc.* the ESKF sustains accuracy and consistency far better than a classical Kalman filter.

Nevertheless, fuzzy logic, in particular, can help with adaptively adjusting the critical Kalman filter parameters (Q and R), which are frequently challenging to estimate under practical circumstances. Several strategies are suggested in literature to enhance

performance when there are uncertainties such as model errors, outliers, and fluctuating noise:

- update Q and R dynamically using adaptive algorithms (*Kumar & Mondal, 2023*).
- Use fuzzy logic to track filter performance and modify settings as necessary (*Al-Ghossini et al., 2016*).
- Make use of robust filter types, such as the H-infinity filter (*Duran-Martin et al., 2024*).
- Prior to filtering, use outlier detection to clean the measurement data (*Navon & Bobrovsky, 2021*).

## PROPOSED METHOD

### Data collection

For the proposed Fuzzy Logic Correlation Scheme, the X-Plane-11 software was used for the aircraft flight simulation. X-Plane is one of the most advanced flight simulators used in industry as well as for research purpose. X-Plane differentiates itself from other simulators by implementing an aerodynamic model called blade element theory (*Yu et al., 2020*; *Gusti Agung Agastya & Kusumoputro, 2019*). Traditionally, flight simulators emulate the real-world performance of an aircraft by using empirical data in predefined lookup tables to determine aerodynamic forces such as lift or drag, which vary with differing flight conditions. The Skywalker X8 model was designed in the X-Plane and was used to generate the flight test data. The sensor data obtained from X-Plane was exported in the MATLAB environment for further processing. The Mission planner software was used to specify the waypoints and fly the mission in the X-Plane as shown in Fig. 2. The Digital Elevation Map (DEM) of the same area was obtained from *NASA Earthdata (n.d)*. An example of the listing of all the eighteen variables of interest generated during the flight simulation is shown in Table 1 below. These variables are the 'inputs' to the proposed navigation scheme.

### INS navigation algorithm and transformations

The Inertial Navigation System is the combination of three major modules, *i.e.*, (1) the inertial measurement unit (IMU) (2) the co-ordinate transformation module and (3) the two-step integrator module. An IMU measures the three Degree of Freedom (DoF) angular rate and magnitudes of specific forces (*Shaukat, Moinuddin & Otero, 2021*). These measurements are initially obtained in the body frame (*i.e.*, the sensor frame) axes. The INS model selected is based on strap-down INS (SINS) technology. In this scheme, the integration is performed using the trapezoidal integration (TI) method to obtain aircraft position. In this work, the east-north-up (ENU) co-ordinate system is selected as the world co-ordinate system. The conversion from ENU frame to the latitude-longitude-altitude (LLA) frame is a two-step process (1) conversion from ENU to Earth-Centered, Earth-Fixed (ECEF) and (2) conversion from the ECEF to LLA frame. The transformation from ECEF to LLA frame can be performed by the non-iterative transformation method defined in *You (2000)* or the close form method called the Jijie Zhu's algorithm (*Osen, 2017*). Moreover, it should be noted that the IMU measures the relative acceleration, also

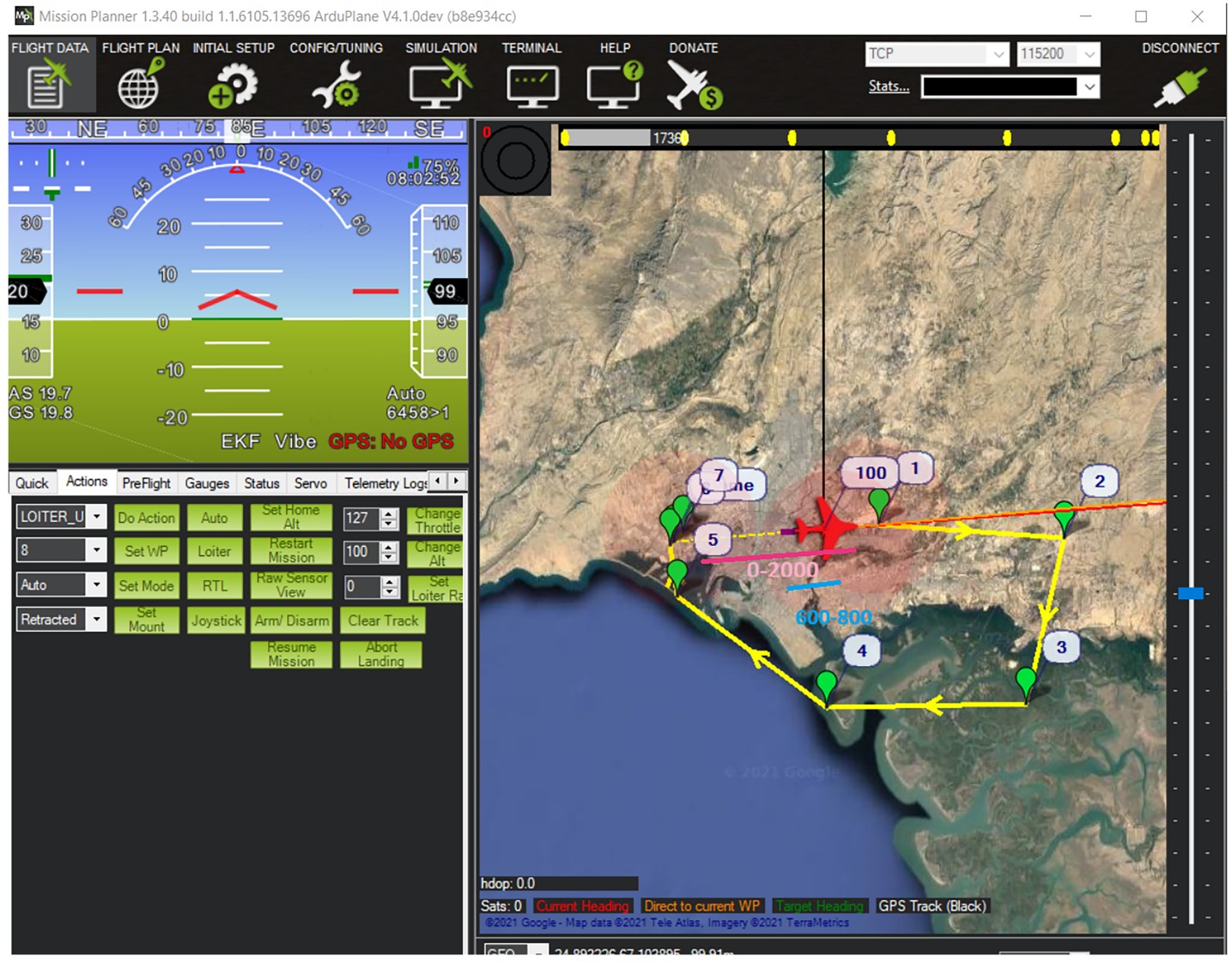

**Figure 2 Snapshot of complete aircraft trajectory for data collection simulated in mission planner software (*Oborne, n.d*).**

**Table 1 The eighteen variables of interest from X-plane simulation.**

| Time (s) | Q (rad/s) | P (rad/s) | R (rad/s) | Pitch (deg.) | Roll (deg.) | Yaw (deg.) | Lat. (deg.) | Long. (deg.) |
|---|---|---|---|---|---|---|---|---|
| 100.8 | −1.01361 | −0.06368 | −0.00787 | 8.47464 | 0.15743 | 43.76252 | 24.88034 | 66.93123 |

| Alt. (m) | Lat. origin (deg.) | Long. origin (deg.) | X (m) | Y (m) | Z (m) | Vx (m/s) | Vy (m/s) | Vz (m/s) |
|---|---|---|---|---|---|---|---|---|
| 32.55061 | 24.50000 | 67.00000 | 6,945.02 | 134.40 | 42,341.21 | 0.00557 | 0.25846 | 0.01509 |

known as the specific force. It does not measure the true kinematic acceleration directly due to presence of the earth's gravitational acceleration effect modelled as:

$$f_i b^b = \dot{R}_i^b \ddot{p}_i - g^b. \tag{1}$$

In the above equation, $\ddot{p}_i^b$ represents acceleration in the inertial frame and $g^b$ represents the gravity in the body frame. $\dot{R}_i^b$ is the rate of change in the rotation matrix represented in the body frame (b). And $f_{ib}^b$ represents the specific force.

### Theoretical analysis of the ESKF

In this section, the convergence of the ESKF is shown to be asymptotically stable by incorporating the result given in *Jazwinski (2007)*, which establishes the asymptotic stability of Kalman filters. The incorporated result is aligned with the structure and assumptions of the ESKF.

Consider the discrete-time nonlinear system:

$$x_{k+1} = f(x_k), \quad \text{where} \quad f : \mathbb{R}^n \to \mathbb{R}^n. \tag{2}$$

A small perturbation $\delta x_k$ around the nominal trajectory $x_k$ evolves as:

$$x_{k+1} + \delta x_{k+1} = f(x_k + \delta x_k). \tag{3}$$

Performing a first-order Taylor series expansion around $x_k$ and neglecting higher-order terms:

$$x_{k+1} + \delta x_{k+1} = f(x_k) + \nabla f(x_k)\delta x_k. \tag{4}$$

Subtracting the nominal dynamics $x_{k+1} = f(x_k)$, we obtain the linearized error dynamics:

$$\delta x_{k+1} = \nabla f(x_k)\delta x_k. \tag{5}$$

Let the state transition matrix be defined as:

$$\Phi_k = I + \Delta t \nabla f(x_k). \tag{6}$$

Then, the discrete-time evolution of the error state becomes:

$$\delta x_{k+1} = \Phi_k \delta x_k. \tag{7}$$

Now consider process noise $\omega_k \sim \mathcal{N}(0, Q_k)$ and measurement noise $v_k \sim \mathcal{N}(0, R_k)$, added as:

$$\delta x_{k+1} = \Phi_k \delta x_k + \omega_k. \tag{8}$$

Assume $z_k$ is the measurement vector and $h(x_k)$ is the nonlinear measurement function. The measurement model is:

$$z_k = h(x_k) + v_k. \tag{9}$$

The Kalman filter (KF) equations for this linearized system are:

Predicted Covariance:

$$P_{k+1}^- = \Phi_k P_k \Phi_k^T + Q_k. \tag{10}$$

Measurement Prediction Covariance:

$$S_{k+1} = H_{k+1} P_{k+1}^- H_{k+1}^T + R_{k+1}. \tag{11}$$

Kalman Gain:

$$K_{k+1} = P_{k+1}^- H_{k+1}^T S_{k+1}^{-1}. \tag{12}$$

Updated Covariance:

$$P_{k+1} = (I - K_{k+1}H_{k+1})P_{k+1}^{-} \tag{13}$$

where

$$H_k = \frac{\partial h(x)}{\partial x}\Big|_{x=\hat{x}_k^{-}} \tag{14}$$

and $\hat{x}_k^{-}$ is the predicted state before update at time step $k$.

*If the system with transition matrix $\Phi_k$ is SCHUR-stable, uniformly completely observable, and $x_k$, $\omega_k$, and $v_k$ are mutually independent, with $P_0 \geq 0$, and both $Q_k$ and $R_k$ bounded, then the error state Kalman filter is uniformly asymptotically stable.*

### The ESKF equations

The error model utilized in the error state Kalman filter (ESKF) is obtained by employing the small perturbation theory (*Rosario et al., 2018*). Similar to the KF, the error state Kalman filter (ESKF) operates in three stages: (1) the initialization stage, (2) the prediction stage and (3) the update stage.

The initialization stage: in the initialization phase, the state vector, process and measurement noise covariances are initialized:

$x_0^{-}$ = initializing the state variables.

$P_0^{-}$ = initialization of the error covariance.

$Q_0$ = initialization of the process noise covariance.

$R_0$ = initialization of the measurement noise covariance.

The prediction stage: in the prediction stage, the error state and the error state covariance matrix is updated. In the ESKF, the error state '$\delta x_k$' is predicted according to the following equation (*Youn & Gadsden, 2019*):

$$\delta x_k^{-} = \Phi_{k-1}\delta x_{k-1} + \tilde{w}_{k-1}. \tag{15}$$

In the above equation $\Phi_{k-1}$, and $\tilde{w}_{k-1}$ represent the state transition matrix and discretized process noise respectively. The error state covariance is propagated using the following equation:

$$P_k^{-} = \Phi_{k-1}P_{k-1}\Phi_{k-1}^{T} + Q_{k-1}. \tag{16}$$

Here $Q_{k-1}$ is the process noise covariance.

The update/correction stage: in the update stage, the sensor measurements are employed to correct the predicted states. In this stage, the residual measurement is updated. It is defined as the difference between the measurements and the prediction of measurements.

$$\delta z_k = z_k - h(\hat{x}_k^{-}). \tag{17}$$

The Kalman gain is then computed as:

$$S_k = R_k + H_k P_k^{-} H_k^{T} \tag{18}$$

$$K_k = P_k^{-} H_k^{T} S_k^{-1} \tag{19}$$

where $S_k$ is the origination vector covariance matrix. It defines the uncertainty associated with the innovation. In the above equation, the matrix 'H' is computed with respect to the error state as:

$$H_k \underset{=}{\Delta} q \frac{\partial h}{\partial \delta x}\Big|_{x=\hat{x}_k^-}. \tag{20}$$

Finally, the innovation matrix is calculated according to the following equation:

$$I_k = \delta z_k - H_k \delta x_k^-. \tag{21}$$

The innovation is the difference between the error of observations and the expected error $H_k \delta x_k^-$. The states are corrected according to:

$$\delta \hat{x}_k = K_k I_k. \tag{22}$$

Since, an ESKF framework is employed in this work, using the error states or the indirect form of the state vector, the values of observation is obtained by the subtraction of the INS based measurements and the aiding sensor in general. In this work, the observation is obtained as the difference between the INS and the terrain based measurements.

$$\delta z = p_{\text{INS}} - p_{\text{TERCOM}}. \tag{23}$$

The posteriori state covariance $P_k^+$ is computed as:

$$P_k^+ = (I - K_k H_k) P_k^-. \tag{24}$$

True State Estimate: the true state is now estimated according to the following equation as the sum of nominal state and the predicted state:

$$\hat{x}_k = \hat{x}_k^- + \delta \hat{x}_k. \tag{25}$$

The above expression is sometimes written as:

$$x = \hat{x} \otimes \delta x \tag{26}$$

where $\otimes$ represents addition operator.

### Design of proposed ESKF filter

In order to generate realistic estimates from the ESKF, the system dynamics must be well modeled. In this work, the ESKF is designed as a 15-state Kalman filter. The error-state vector is divided into position, velocity, acceleration (PVA) and sensor biases. This split improves clarity and supports better modeling. It separates fast-changing states (PVA) from slow or hidden states (sensor biases). This grouping matches how real systems behave and helps track changes more clearly. PVA states are often easier to observe. Sensor biases, in contrast, change slowly and are harder to track. This structure also matches standard EKF-based navigation setups. In such filters, sensor biases are often modeled as random walks, while PVA states follow motion dynamics. By following this structure, we stay in line with proven approaches in navigation and estimation.

$$\mathbf{x} = \left[\delta\phi, \delta\lambda, \delta h, \delta V_N, \delta V_E, \delta V_D, \dot{\phi}_N, \dot{\phi}_E, \dot{\phi}_D, \omega_N, \omega_E, \omega_D, B_N, B_E, B_D\right]^T \tag{27}$$

$$\dot{\mathbf{x}} = \mathbf{Ax} \tag{28}$$

$$
\begin{bmatrix}
\delta\dot{\phi} \\
\delta\dot{\lambda} \\
\delta\dot{h} \\
\delta\dot{V}_N \\
\delta\dot{V}_E \\
\delta\dot{V}_D \\
\dot{\phi}_N \\
\dot{\phi}_E \\
\dot{\phi}_D \\
\dot{\omega}_N^{dr} \\
\dot{\omega}_E^{dr} \\
\dot{\omega}_D^{dr} \\
\dot{B}_N \\
\dot{B}_E \\
\dot{B}_D
\end{bmatrix}
=
\begin{bmatrix}
0 & 0 & A_1 & A_2 & 0 & 0 & 0 & 0 & 0 & 0 \\
A_3 & 0 & A_4 & 0 & A_5 & 0 & 0 & 0 & 0 & 0 \\
A_3 & 0 & 0 & 0 & 0 & -1 & 0 & 0 & 0 & 0 \\
A_6 & 0 & A_7 & A_8 & A_9 & \phi & -f_D & f_E & 0 & 0 \\
A_{10} & 0 & A_{11} & A_{12} & A_{13} & A_{14} & f_D & -f_E & 0 & 0 \\
A_{15} & 0 & A_{16} & -2\phi & A_{17} & 0 & -f_E & f_N & 0 & 0 \\
A_{18} & 0 & A_{19} & A_{20} & 0 & A_{21} & \phi & -1 & 0 & 0 \\
0 & 0 & A_{22} & A_{23} & 0 & A_{24} & A_{25} & 0 & -1 & 0 \\
A_{26} & 0 & A_{27} & A_{28} & -\phi & A_{29} & 0 & -\beta & -1 & 0 \\
0 & 0 & 0 & 0 & 0 & 0 & 0 & 0 & -\beta & 0 \\
0 & 0 & 0 & 0 & 0 & 0 & 0 & 0 & -\beta & 0 \\
0 & 0 & 0 & 0 & 0 & 0 & 0 & 0 & -\beta & 0 \\
0 & 0 & 0 & 0 & 0 & 0 & 0 & 0 & 0 & -\beta \\
-\beta & 0 & 0 & 0 & 0 & 0 & 0 & 0 & 0 & 0 \\
0 & -\beta & 0 & 0 & 0 & 0 & 0 & 0 & 0 & 0
\end{bmatrix}
\begin{bmatrix}
\delta\phi \\
\delta\lambda \\
\delta h \\
\delta V_N \\
\delta V_E \\
\delta V_D \\
\phi_N \\
\phi_E \\
\phi_D \\
\omega_N^{dr} \\
\omega_E^{dr} \\
\omega_D^{dr} \\
B_N \\
B_E \\
B_D
\end{bmatrix}
\tag{29}
$$

$$A_1 = -\frac{\phi}{R_m + h}, \quad A_2 = \frac{1}{R_m + h}, \quad A_3 = \dot{\lambda}\,tg\phi, \quad A_4 = \frac{-\dot{\lambda}}{R_p + h}, \quad A_5 = \frac{1}{(R_p + h)\cos\phi},$$

$$A_6 = V_E \cos\phi\left[2\omega_e + \dot{\lambda}\sec^2\phi\right], \quad A_7 = \left[\frac{V_E\dot{\lambda}\sin\phi}{R_p + h} - \frac{V_D\dot{\phi}}{R_M + h}\right], \quad A_8 = \frac{V_D}{R_m + h},$$

$$A_9 = -2(\omega_e + \dot{\lambda})\sin\phi, \quad A_{10} = \left[2\omega_e(V_N \cos\phi - V_D \sin\phi) + \dot{\lambda}V_N \sec\phi\right],$$

$$A_{11} = \frac{-\dot{\lambda}}{R_p + h}[V_D \cos\phi + V_N \sin\phi], \quad A_{12} = (2\omega_e + \dot{\lambda})\sin\phi, \quad A_{13} = \frac{1}{R_p + h}[V_D + V_N\,tg\phi],$$

$$A_{14} = (2\omega_e + \dot{\lambda})\cos\phi, \quad A_{15} = 2\omega_e V_E \sin\phi, \quad A_{16} = \left[\frac{V_N}{R_M + h}\dot{\phi} + \frac{V_E}{R_p + h}\dot{\lambda}\cos\phi + (k-2)\frac{g}{R_e}\right],$$

$$A_{17} = -2(\omega_e + \dot{\lambda})\cos\phi, \quad A_{18} = -\omega_e \sin\phi, \quad A_{19} = \frac{-\dot{\lambda}}{R_p + h}\cos\phi, \quad A_{20} = \frac{1}{R_p + h}, \quad A_{21} = -(\omega_e + \dot{\lambda})\sin\phi,$$

$$A_{22} = \frac{\dot{\phi}}{R_M + h}, \quad A_{23} = -\frac{1}{R_M + h}, \quad A_{24} = (\omega_e + \dot{\lambda})\sin\phi; \quad A_{25} = (\omega_e + \dot{\lambda})\sin\phi;$$

$$A_{26} = -\left(\omega_e \cos\phi + \dot{\lambda}\sec\phi\right), \quad A_{27} = \frac{-\dot{\lambda}}{R_p + h}\sin\phi, \quad A_{28} = \frac{-tg}{R_g + h}\phi, \quad A_{29} = -(\omega + \dot{\lambda})\cos\phi \tag{30}$$

**Table 2 Description of important variables.**

| Symbols | Description |
|---|---|
| $\varphi$ | Position co-ordinate—Latitude |
| $\lambda$ | Position co-ordinate—Longitude |
| $h$ | Position co-ordinate—Height |
| $V_e$ | Velocity (East) |
| $V_n$ | Velocity (North) |
| $V_d$ | Velocity (Down) |
| $R_L$ | Radii of curvature along meridian |
| $t$ | Time |
| $g$ | Gravitational acceleration |
| $R_E$ | Radii of curvature in parallel |
| $\omega$ | Angular velocity of the earth's rotation |
| $f_n$ | Specific force (North) |
| $f_e$ | Specific force (East) |
| $f_d$ | Specific force (Down) |
| $k$ | Gain co-efficient |
| $R$ | Radius of the earth |
| $\beta$ | Shaping filter coefficient (Correlation coefficient) |

where, the values mathematically describing equations $A_1 - A_{29}$ are all scalars. The description of variables in above equations are defined in Table 2. The values of the constant factors in the system matrix are given as: $R_m = 6335.439 \times 10^3$, $g = 9.8$, $R_e = 6371.0072 \times 10^3$, $R_p = 6399.594 \times 10^3$ and $\omega_e = 7.3 \times 10^{-5}$

The innovation is computed as:

$$I_k = \delta z_k - H_k \delta x_k^- = \begin{bmatrix} L_{INS} - L_{TERCOM} \\ l_{INS} - l_{TERCOM} \\ h_{INS} - h_{BAROMETER} \end{bmatrix} - H \begin{bmatrix} \delta f \\ \delta \lambda \\ \delta v_N \\ \delta v_E \\ \delta v_D \\ \dot{\phi}_N \\ \dot{\phi}_E \\ \phi_D \\ \omega_N \\ \omega_E \\ \omega_D \\ \omega_u \\ \omega_v \\ \omega_w \\ B_N \\ B_E \\ B_D \end{bmatrix}. \tag{31}$$

The transformation matrix 'H' is given by the following:

$$H = \begin{bmatrix} I_{3\times3} & 0_{3\times3} & 0_{3\times3} & 0_{3\times3} & 0_{3\times3} \end{bmatrix}. \tag{32}$$

The measurement co-variance R is set to:

$$R = \begin{bmatrix} \sigma_x & 0 & 0 \\ 0 & \sigma_y & 0 \\ 0 & 0 & \sigma_z \end{bmatrix}. \tag{33}$$

For the prediction step of the ESKF to yield meaningful results, it must rely on an accurate ground truth. This ground truth can be obtained using the TERCOM scheme, where the reference location $L_{\text{TERCOM}}$ and the location $l_{\text{TERCOM}}$ are derived by correlating the onboard altitude data with the digital elevation model (DEM), typically using algorithms such as the mean absolute deviation (MAD) method. The following section elaborates on the challenges associated with obtaining this ground truth.

## DEM processing and conventional correlation scheme

In the proposed framework, the Mean Absolute Deviation (MAD) algorithm is used for the purpose of matching, where the matching area selection is based on Fuzzy Logic approach detailed in the next section. The MAD algorithm correlates the terrain heights estimated from the system with the DEM based topographical values. This correlation is described by the following equation:

$$MAD_{n,m} = -\frac{1}{N}\sum_{i=1}^{N}\left|h_{meas}^{(i)} - h_{DB}(n+i, m)\right|. \tag{34}$$

In the above equation, $h_{meas}$ and $h_{DB}$ are representing the measured terrain heights and the stored terrain database respectively. The variable 'N' represents the number of samples. Also, n and m represent the rows and columns of the terrain matrix stored in the digital terrain database. The position fix is obtained using the minimum MAD value given by:

$$P_{MAD}(n, m) = P(\arg\min_{n,m}(MAD_{n,m})). \tag{35}$$

Applying the correlation process, the terrain height with the minimum values are considered as appropriate candidates. The steps comprise of comparing and matching the difference between the barometer and the radar altimeter measurements (which is essentially the terrain height) with a pre-stored DEM. The DEM models topography as well as the topographical change in a geographical region (*Wilson & Ramirez-Serrano, 2014*). The DEM is structured as a two-dimensional matrices of elevation values assigned as the DEM matrix Z. In a DEM cell, the geographical location is computed according to a specific co-ordinate system assigning a projected location in the two-dimensional XY plane. The terrain roughness and uniqueness factor is also crucial for the successful application of the TERCOM system. TERCOM results are more appreciable for unique terrains (*Chen, Xu & Ding, 2022*). The terrain roughness characteristics of the region for which the simulations were conducted are defined by the two parameters, *i.e.*, the Sigma-Z and Sigma-T values. These parameters are employed as the terrain roughness indicators

defined in *Lu, Jian & Xiaowen (2019)*, *Raković, Simonović & Grbović (2020)*. The terrain roughness described by Sigma-T is represented by the standard deviation of terrain elevation values. Areas with very flat terrains and lakes have smaller Sigma-T values. Similar to Sigma-T values, Sigma-Z represents the roughness in the terrain. Another parameter is the terrain correlation length. The terrain correlation length describes the separation between two rows or columns of the terrain matrix which are required to reduce their normalized auto-correlation function to $e-1 = 0.3697$. In this article, only the first two parameters, *i.e.*, Sigma-T and Sigma-Z are adopted for the analysis of terrain roughness characteristics.

## Fuzzy logic based ROI selection for TERCOM

The conventional mean absolute devotion (MAD) algorithm is found to be computationally expensive in the traditional TERCOM-based strategy. In the proposed method, a heuristic based fuzzy logic system (FLS) has been designed to select the matching area in order to improve the shortcomings of the conventional MAD algorithm. The matching area selection is performed using the aircraft heading and roll angle parameters as inputs to the FLS. Since in practical situations, not only the yaw angle can change during the actual flight, the roll angle may also vary which affects the matching region due to aircraft maneuver.

The FLS module calculates the co-ordinates of the rectangular region (defining the matching area) based on the instantaneous heading angle (Y) and roll angle (R) information from the aircraft orientation sensor. The proposed fuzzy logic system takes the yaw (Y) and roll angle (R) as inputs and provides the co-ordinates (E1, E2, N1, N2) that define the rectangular region of the matching area based on the established fuzzy rules. The inputs and output membership functions are selected to be triangular membership functions. Because of their simplicity, effectiveness, and speed of computation, triangular membership functions are suited for real-time systems with constrained resources (*Varshney & Goyal, 2023*). They are simple to interpret and tune due to their intuitive design and low memory requirements. Control and decision-making tasks can benefit from these functions' ability to accurately approximate gradual transitions between fuzzy sets and their compatibility with fuzzy rule-based systems due to their simplicity (*Khairuddin et al., 2021*; *Samonto et al., 2021*). The FLS system is designed as a 2-input and 4-output multi-input-multi-output (MIMO) system. Figure 3 shows the relationship between the inputs variables (Y and R) and output variables (E1, E2, N1 and N2). For the proposed system, the domain of the yaw angle parameter is considered from [0, 360] and the aircraft roll angle is considered within the domain [−45, 45]. In Fig. 3, the four regions for aircraft maneuver are based on the aircraft heading angle parameter. For each region, a separate FLS module is designed and stored in the system. In Fig. 3 the Region-I of the aircraft motion is defined from 0 to 90 degrees with respect to the yaw angle. The input membership functions for this case are shown in Fig. 4. In Fig. 3, the parameter 'L' represents a parameter related to the terrain characteristics over which the aircraft is flying. For simplification, this parameter is set to a constant parameter in this article. Based on parameter L and yaw and $\theta$, the computation of the output variables E1, E2, N1 and N2 is

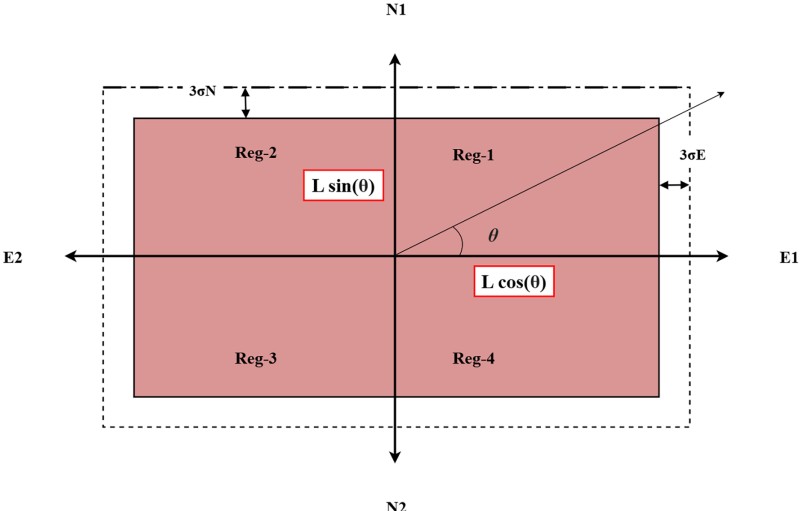

**Figure 3  Schematic of the matching area selection.**

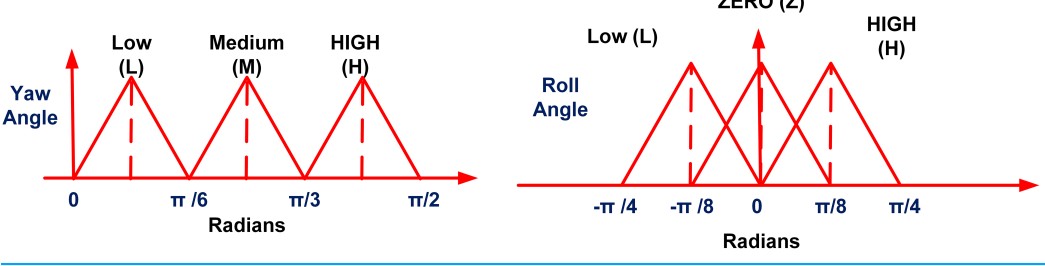

**Figure 4  Membership functions for inputs.**

possible, however, the uncertainty arises when there are changes in the roll angle parameter as well as sensor noise parameters. Therefore, while constructing the output membership functions (MFs), the domain of outputs is also defined considering the errors in the north and east directions, *i.e.*, $\sigma_E$ and $\sigma_N$ respectively. The output membership functions are shown in Figs. 5 and 6 respectively. The output membership functions are designed incorporating the possible changes in position in the East and North frame due to the effect of roll angle parameter. With reference to Fig. 3, the description of the output fuzzy variables is provided in Table 3. The corresponding set of fuzzy rules are provided in Table 4. The fuzzy rules for the remaining three regions are constructed in a similar manner. According to Fig. 3, the rules are constructed according to the following logic:

*R1: IF the yaw angle is small and roll angle is zero THEN set E1 and E2 high and N1 and N2 low*

With reference to the above logic, we define the input subset of three-degree division, $S_{in1} = \{Low, Medium, High\}$ for input-1 (yaw angle) and $S_{in2} = \{Low, Medium, High\}$ for input-2 (*i.e.*, the roll angle corresponding to a change from negative to positive).

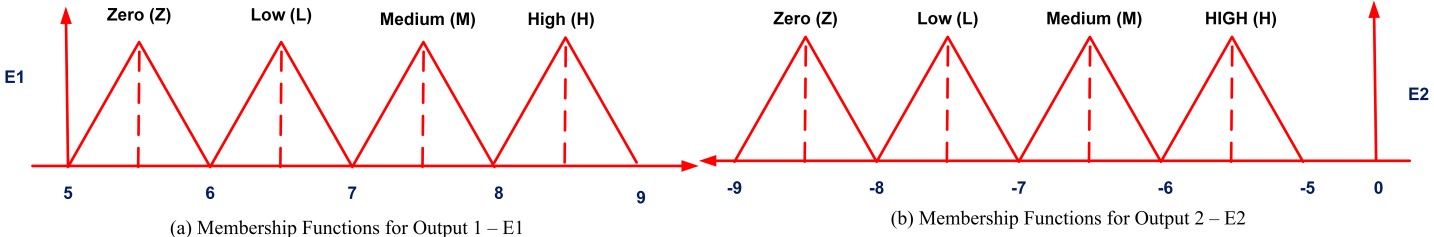

(a) Membership Functions for Output 1 – E1

(b) Membership Functions for Output 2 – E2

**Figure 5 Membership functions for outputs 1 and 2.**

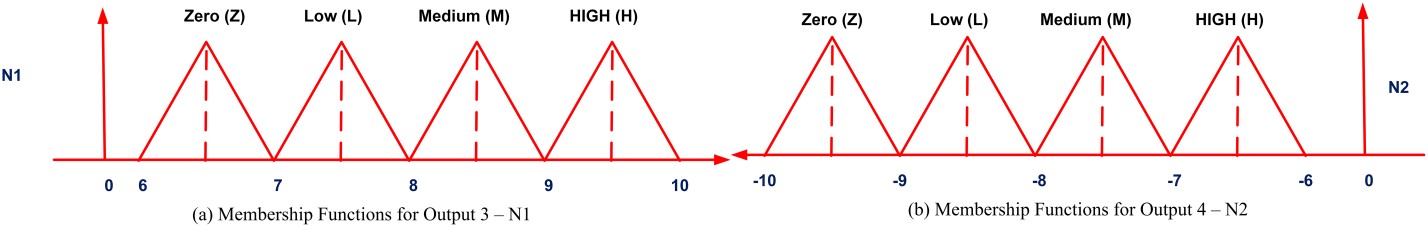

(a) Membership Functions for Output 3 – N1

(b) Membership Functions for Output 4 – N2

**Figure 6 Membership functions for outputs 3 and 4.**

| Table 3 Description of fuzzy output variables. | |
| --- | --- |
| **Parameter** | **Description** |
| E1 | West boundary of the rectangular region |
| E2 | East boundary of the rectangular region |
| N1 | North boundary of the rectangular region |
| N2 | South boundary of the rectangular region |

| Table 4 Fuzzy rule base—region-I. | |
| --- | --- |
| **Rule** | **Description** |
| R1: | If Y is L and R is Z, then E1 is H and E2 is H and N1 is L and N2 is L. |
| R2: | If Y is M and R is Z, then E1 is M and E2 is M and N1 is M and N2 is M. |
| R3: | If Y is H and R is Z, then E1 is L, E2 is L, N1 is H and N2 is H. |
| R4: | If Y is L and R is L, then E1 is M and E2 is M and N1 is L and N2 is L. |
| R5: | If Y is M and R is L, then E1 is M and E2 is M and N1 is M and N2 is M. |
| R6: | If Y is H and R is L, then E1 is L, E2 is L, N1 is H and N2 is H. |
| R7: | If Y is L and R is H, then E1 is M and E2 is M, N1 is L and N2 is L. |
| R8: | If Y is M and R is H, then E1 is L and E2 is L and N1 is M and N2 is M. |
| R9: | If Y is H and R is H, then E1 is L and E2 is L and N1 is H and N2 is H. |

The output subset is defined with a four-degree division given by $S_{out} = \{Zero, Low, Medium, High\}$, corresponding to the minimum and maximum values for the rectangular boundaries, *i.e.*, E1, E2, N1, N2.

**Table 5  INS initial parameters for the two test cases.**

| Parameter | Test Case-I | Test Case-II |
|---|---|---|
| Initial latitude | 24.88034° | 33.55415° |
| Initial longitude | 66.93123° | 72.81963° |
| Initial heading angle | 0.15743° | 9.9222° |
| Initial pitch angle | 8.47464° | 0.14436° |
| Initial roll angle | −0.00787° | 7.94772° |
| Sample time of the orientation sensor | 0.12 s | 0.12 s |

## SIMULATION RESULTS

In order to determine the effectiveness of the proposed ESKF algorithm with the designed system matrix along-with the proposed heuristic based fuzzy MAD algorithm, the simulation results are presented in this section. The simulation was conducted using MATLAB software environment. For validation of the proposed method, additionally, the GPS signals, *i.e.*, the latitude and longitude from the X-Plane simulator were also included in the dataset. The tests were conducted for two specific cases with two different DEMs having differences in topographical features. The terrain data used in this research was obtained from the NASA Earthdata Search portal (https://search.earthdata.nasa.gov/search). In particular, Shuttle Radar Topography Mission (SRTM) and ASTER GDEM are two publicly accessible datasets from which digital elevation model (DEM) data was obtained. The spatial resolution of these datasets is usually 30 m (1 arc-s) worldwide, though some regions are also accessible at 90 m (3 arc-s). For research and educational purposes, the data is freely accessible, making it an affordable resource for terrain-based navigation applications such as TERCOM. In Test-I, the aircraft completes its flight for a total duration of 6,403 s with different altitudes, heading angles and velocity ranges. From this trajectory, the aircraft motion for the time period of 200 s, *i.e.*, from t = 600 to 800 s was considered for the application of the proposed method as shown in Fig. 2. Similarly, for Test-II, the aircraft completes its flight in the total time of 18,000.8 s. For this case, aircraft motion from time t = 12,100–12,200 s was considered for testing the efficacy of the proposed method.

In the X-plane simulation, the radar altimeter on the aircraft registers the height measurement after every 0.8 s. From the complete aircraft motion, the aircraft trajectory and the corresponding motion dynamics are considered for the two specific cases. The INS data was also obtained from the X-plane consisting of the three dimensional velocity vector: $\mathbf{V} = [V_x \ V_y \ V_z]$ and the aircraft orientation angles, *i.e.*, the roll, pitch and yaw $[\psi, \phi, \theta]$ angles. The simulation parameters for the two test cases are given in Table 5 below:

From the complete DEM of dimension 3,600 × 3,600, the region-of-interest (ROI) was extracted having dimension 1,956 × 635 for Test Case I. For the ROI for Test case II, the DEM sub-region selected was of dimensions 1,500 × 1,800. The aircraft trajectory and

**Table 6 Selection of the P, Q, and R matrices.**

| Matrix | Description | Values |
|---|---|---|
| P (Test Case-I & II) | Error covariance | $\text{diag}([10.02]_{15 \times 15})$ |
| Q (Test Case-I & II) | Process noise covariance | $\text{diag}([0.5]_{15 \times 15})$ |
| R (Test Case-I) | Measurement noise covariance | $\text{diag}([50\ 50\ 200]_{3 \times 3})$ |
| R (Test Case-II) | Measurement noise covariance | $\text{diag}([150\ 100\ 200]_{3 \times 3})$ |

**Table 7 Values of the adjustable variables.**

| Variable | Description | Values |
|---|---|---|
| $\beta$ | Filter shaping coefficient | 0.1 |
| $k$ | Gain coefficient | 4 |

flight dynamics were considered for Test Case I from t = 600–800 s and for Test Case II from t = 12,000–12,200 s. The errors in the positions were corrected using the improved MAD correlation algorithm. The initial position vector, for Test Case-I and for Test Case-II was provided to the strap-down INS system. $(\text{lat}_{\text{initial}}, \text{lon}_{\text{initial}}) = [24.88035°\ 66.93124°]$, $(\text{lat}_{\text{initial}}, \text{lon}_{\text{initial}}) = [33.56°\ 72.82°]$.

The gyroscope mounted on the aircraft is used to measure the roll, pitch and the yaw orientation angles. The robustness of the position estimates generated from the proposed method is determined using the mean squared error (MSE), the mean absolute error (MAE) and the root mean square error (RMSE) metrics.

In the ESKF, the selection of the error covariance (P), process noise covariance (Q) and the measurement noise covariance (R) is described in Table 6. For the proper functioning of the ESKF algorithm, the tuning of the P, Q and R matrices is essential to train the algorithm. The tuning of these matrices is based on the features in the sensor noise and error characteristics. In this work, this tuning is accomplished using the trial and error method after extensive simulation. The 'R' matrix controls the errors in the sensor measurements given as input to the ESKF. The 'Q' matrix controls the variations in the output estimates. The values of the process covariance matrix P which is a diagonal matrix of dimension 15 × 15 is set to '0.5'. For matrix Q, the values are arranged in the form of a diagonal matrix. The first three values in the diagonal are set to be 1e−3, 2e−3 and 12 respectively. The values of the measurement noise covariance matrix for Test Case-I is R = [50 50 200] and for Test Case-II, R = [150 100 200]. The change in R matrix is due to the change in the DEM region in Test Case-II. Furthermore, the values of the adjustable variables in the system matrix, $\beta$ and k for the two cases was selected as defined in Table 7.

Furthermore, the terrain roughness and uniqueness factor is also crucial for the successful application of the TERCOM system. TERCOM results are more appreciable for unique terrains (*Lu, Jian & Xiaowen, 2019*). The terrain roughness characteristics of the region for which the simulations were conducted are defined by the two parameters, *i.e.*,

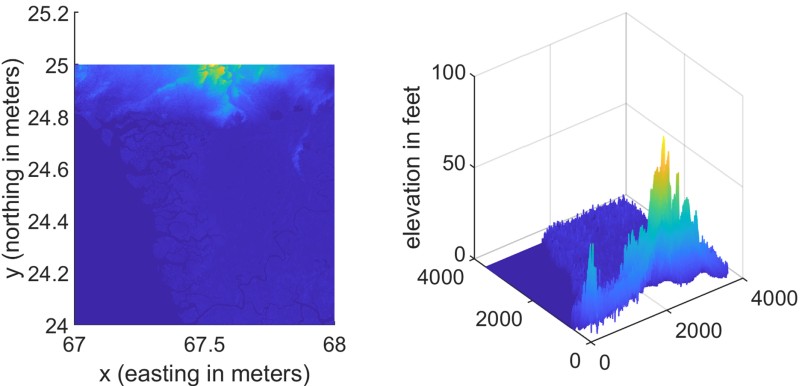

**Figure 7 Terrain profile and the surface terrain plots for Test Case I.**

the Sigma-Z and Sigma-T values. These parameters are employed as the terrain roughness indicators (*Raković, Simonović & Grbović, 2020*). The two terrain parameters Sigma-T and Sigma-Z are given as:

$$\sigma_T = \sqrt{\frac{1}{N}\sum_{i=1}^{N}(\mathbf{H}_i - \bar{\mathbf{H}})^2} \tag{36}$$

$$\bar{\mathbf{H}} = \left(\frac{1}{N}\right)\sum_{i=1}^{N}\mathbf{H}_i; \tag{37}$$

$$\sigma_Z = \sqrt{\frac{1}{N}\sum_{i=1}^{N}(\mathbf{D}_i)^2} \tag{38}$$

$$\mathbf{D}_i = \mathbf{H}_i - \mathbf{H}_{i+1} \tag{39}$$

$$\mathbf{D} = \frac{1}{N-1}\sum_{i=1}^{N-1}\mathbf{D}_i. \tag{40}$$

## Test Case-I

For the first test case the DEM having dimensions of $1{,}500 \times 635$ is shown in Fig. 7 where the bright colors in the contour plot represent higher altitudes. The corresponding three dimensional plot has also been shown in Fig. 7, where the brightest region (*i.e.*, the yellow color) represents the highest elevation values. The region of complete aircraft maneuver is shown in Fig. 8 in red color in the region defined by latitude between 24.7 to 24.9 degrees and the longitudes between 66.9 to 67.4 degrees. In Fig. 8, the aircraft starts its flight initially from the point $= [24.88°, 66.93°]$, continues its flight to the point $= [24.96°, 66.97°]$ and returns back towards its initial point. The flight path between these position values are selected as the region of interest, *i.e.*, the ROI which is highlighted by the ellipse in Fig. 8 and the magnified region is shown in Fig. 8 where the green color represents the region where the testing using the modified MAD and ESKF algorithm was performed. In Fig. 9, it was observed that in this region, the latitude and longitude of the

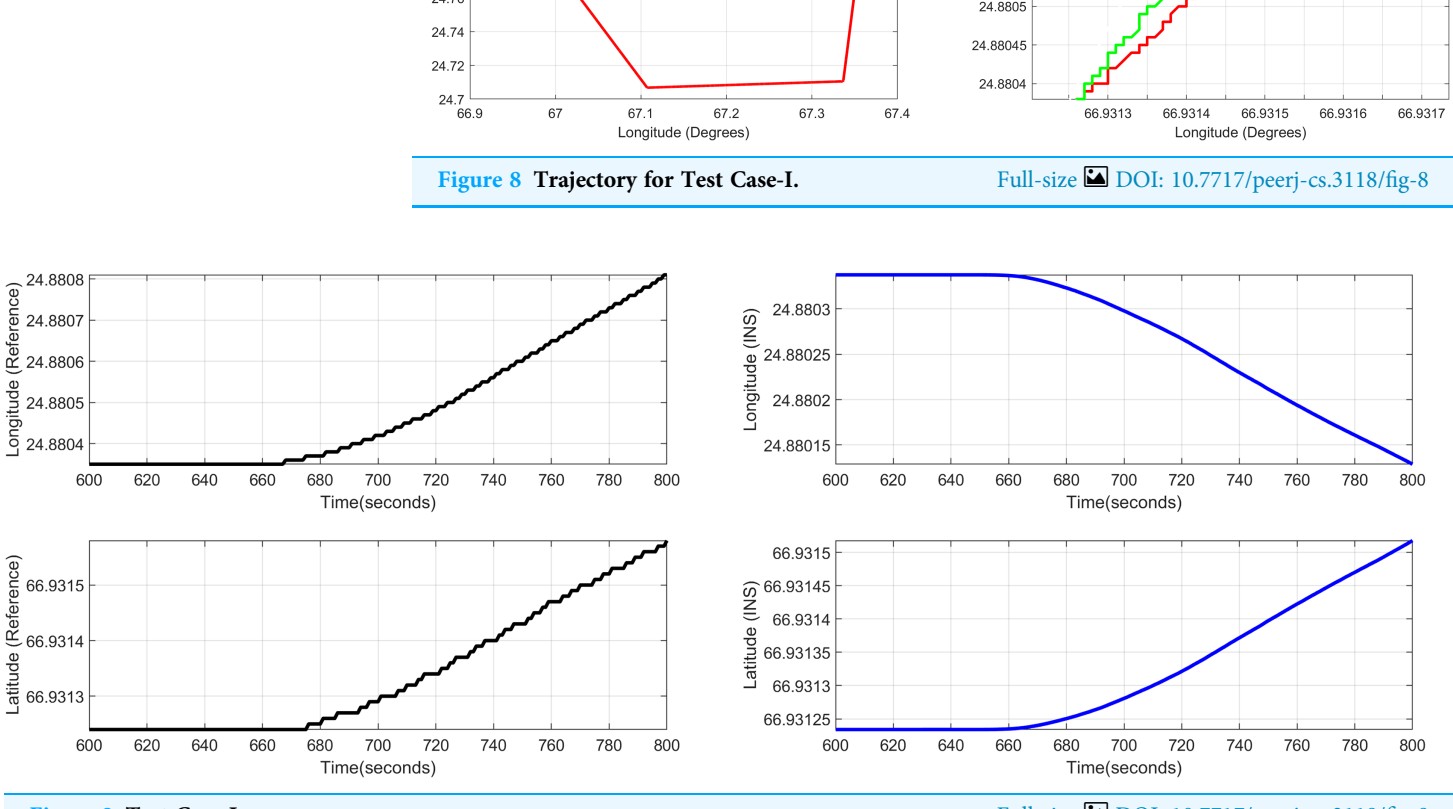

**Figure 8 Trajectory for Test Case-I.**

**Figure 9 Test Case-I.**

aircraft were both increasing. The GPS measurements are used as reference for comparing the position obtained from the proposed method. In Fig. 9, the latitude and longitude obtained using the INS are shown, in which the drifts can be observed in the INS position estimates, *i.e.*, the negative drift error in the longitude position generated by the INS. This affects the position information of the overall trajectory followed by the aircraft. In Fig. 10, the ESKF prediction in the test region are shown, the reference positions (*i.e.*, the latitude and longitude) are represented by blue color and the estimated positions are shown in red. In Fig. 11 the variation in the aircraft velocity and 3D acceleration components are shown respectively. Figure 11 shows that the aircraft velocity increases from the time t = 600–800 s with the maximum velocity being 6 m/s. The aircraft yaw angle and the roll angle variations for Test-Case-I are shown in Fig. 11 respectively. The yaw angle follows a decreasing trend from 0.62 radians to 0.52 radians, whereas the roll angle can be considered approximately 0. In the proposed FL based MAD scheme, this corresponds to the case where only one angle, *i.e.*, the yaw angle of the aircraft is changing and the effect of

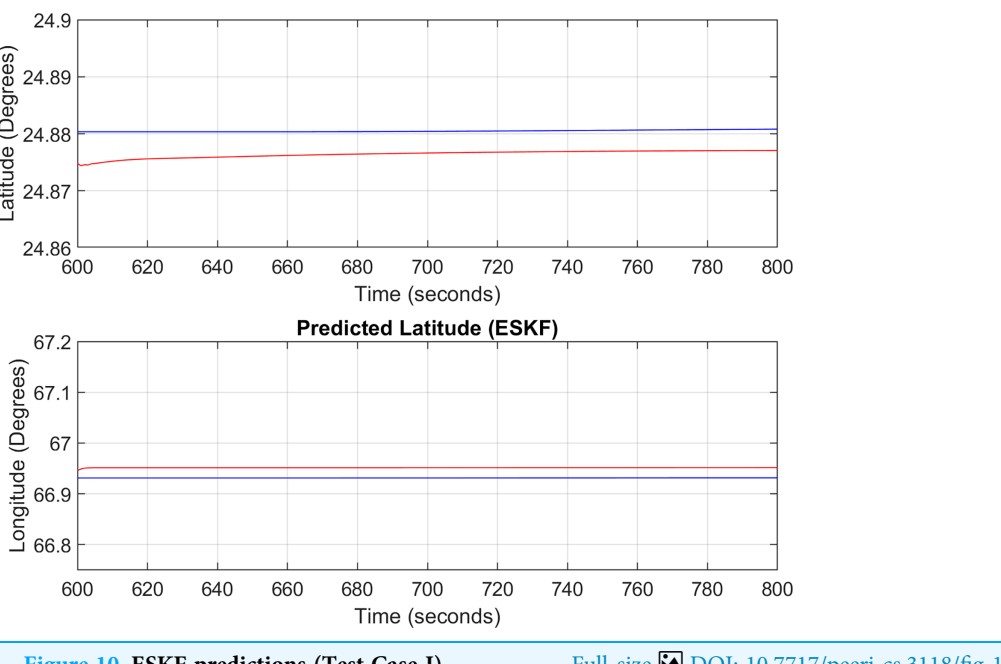

**Figure 10 ESKF predictions (Test Case-I).**

roll angle is not being considered. The ESKF prediction error is shown in Fig. 12, where the error range in the final prediction are 0.02–0.096 degrees respectively showing the effectiveness of the system matrix design of the proposed ESKF method and the optimal tuning of the filter parameters.

## Test Case II

In the second test case the DEM having dimensions of 1,501 × 1,801 is shown in Fig. 13, where the bright colors in the contour plot represent higher altitudes. These values are considered as the terrain topographical database values. The corresponding three dimensional plot is shown in Fig. 13, where the brightest region (*i.e.*, the yellow color) represents the highest elevation. In contrast to the Test Region-I, the region for Test-II is composed of more terrain feature. The region of complete aircraft maneuver is shown in Fig. 14 respectively. The flight path is shown in red color defined by latitude between 33.573 degrees to 33.581 degrees and the longitudes between 72.785–72.835 degrees. The aircraft starts its flight initially from the point = [33.57, 72.79] and continues its flight to the point [33.58, 72.83]. The flight path between these position values are selected as the ROI as shown in Fig. 14 where the region defined in the green color represents the region where the testing using the modified MAD and ESKF algorithm was performed. In Fig. 15, the GPS-based positions for this test case are shown. It can be observed that in this region the latitude of the aircraft is decreasing while the longitude of the aircraft is increasing. For this case, the test data was selected from time t = 12,000–12,200 s. Similar to Test Case-I, the aircraft sensors (*i.e.*, the radar altimeter and the barometer) were used to compare the terrain height feature from the second pre-stored DEM. In Fig. 15, the latitude and longitude obtained using the INS is shown, in which the drifts can be observed in the INS

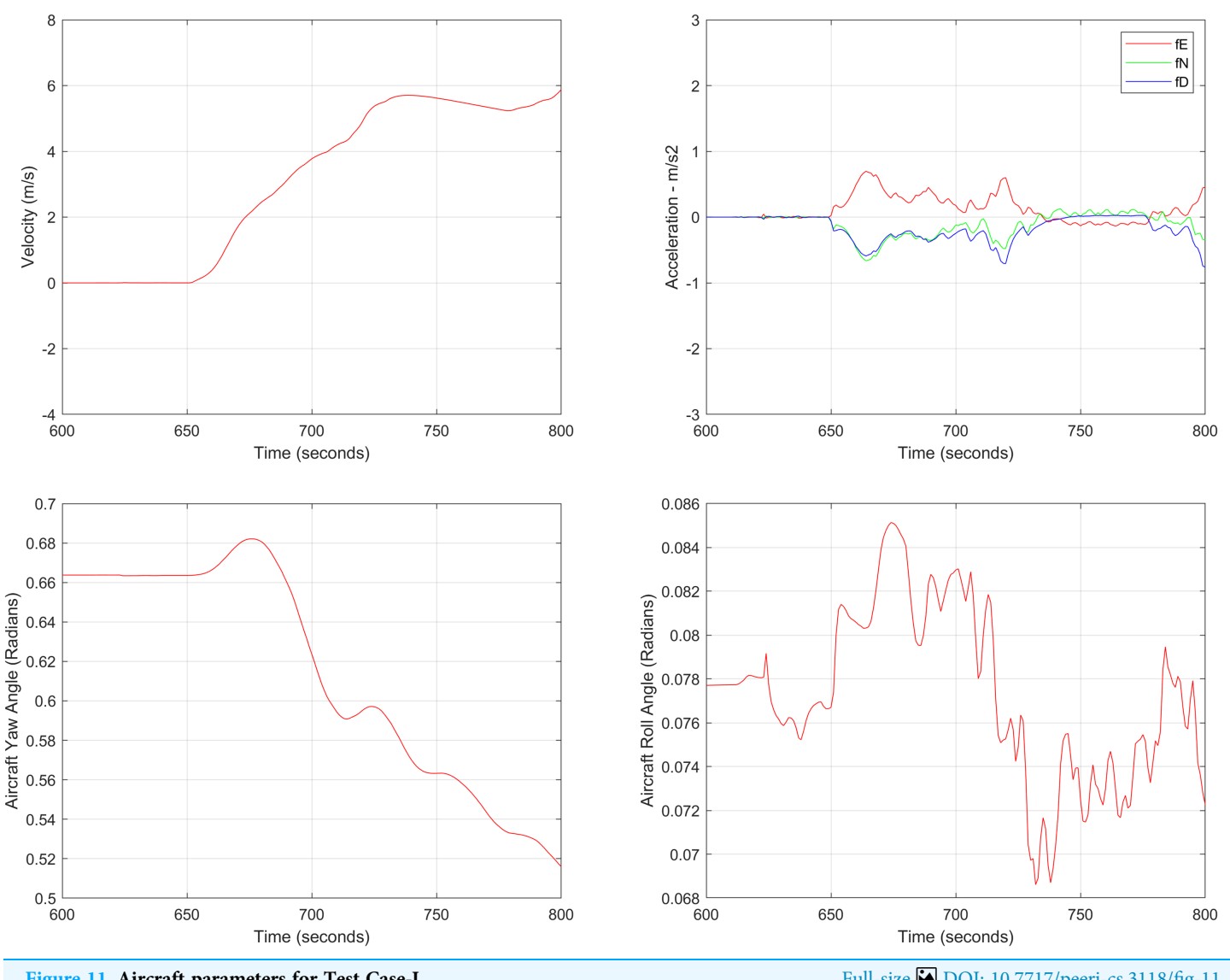

**Figure 11 Aircraft parameters for Test Case-I.**               

position estimates, *i.e.*, compared with the GPS reference position shown in Fig. 15, the negative drift error in the longitude position is generated by the INS which affects the position information of the aircraft. In Fig. 16, the estimates of the position outputs, *i.e.*, the latitude and longitude are shown obtained from the proposed scheme. It can be observed that the drifts are significantly reduced when compared with the INS-based estimations. The velocity and acceleration of the aircraft for this test case are shown in Fig. 17 respectively. The velocity range is from 14.0–14.5 m/s. The velocity remains almost constant for t = 12,000–12,140 s and increases after t = 12,140 s. The velocity of the aircraft remains constant *i.e.*, V = 14.44 m/s seconds for the rest of the aircraft motion from the time t = 12,160–12,200 s. The range of the aircraft heading *i.e.*, yaw during this time is 4.766–4.768 radians and the roll angle is within the range of 0.19–0.22 radians as shown in Fig. 17 respectively. In this test case, the yaw angle variation is such that it remains constant

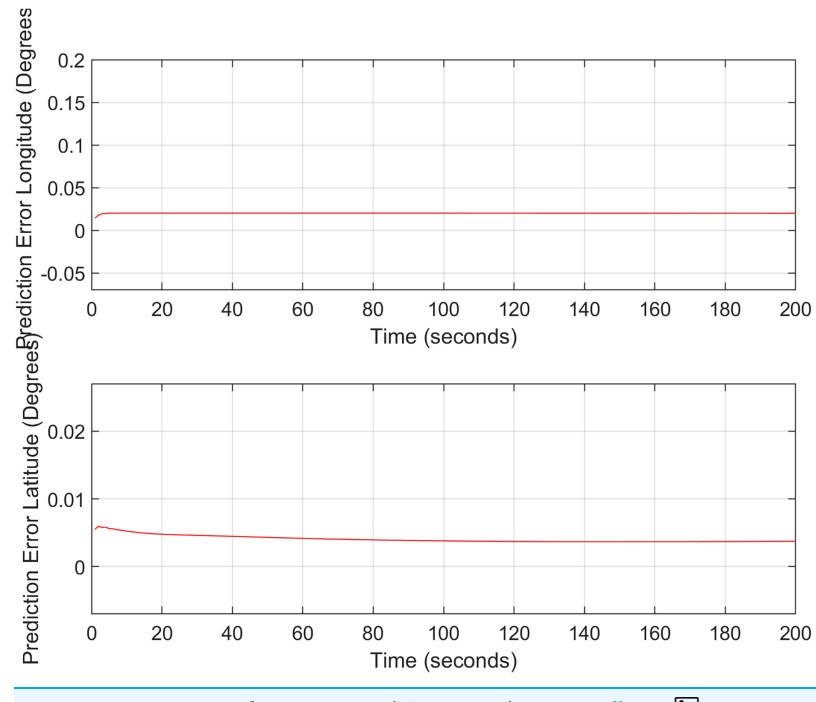

**Figure 12  ESKF prediction errors (Test Case-I).**     

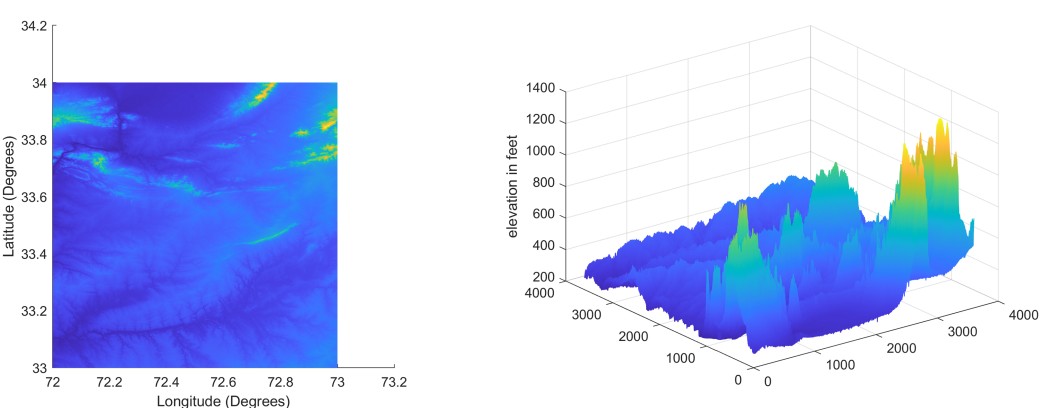

**Figure 13  Terrain profile and the surface terrain plots for Test Case II.**

till t = 12,140 s, increases rapidly after time t = 12,140 s and then decreases. While the roll angle starts decreasing slowly and a rapid change can be observed after time t = 12,140. This corresponds to the case where the effects of variation in both the yaw and the roll angle of the aircraft are considered in the proposed FL based MAD scheme. The predicted errors from the ESKF strategy are shown in Fig. 18. The latitude errors are obtained in the range 0.055–0.5 degrees. While the longitude errors are between 0.098 and 0.1047 degrees depicting minimum error in the acceptable ranges. This highlights the effectiveness of the system matrix design of the proposed ESKF method and the optimal tuning of the filter parameters and the appropriate design of the fuzzy logic system.

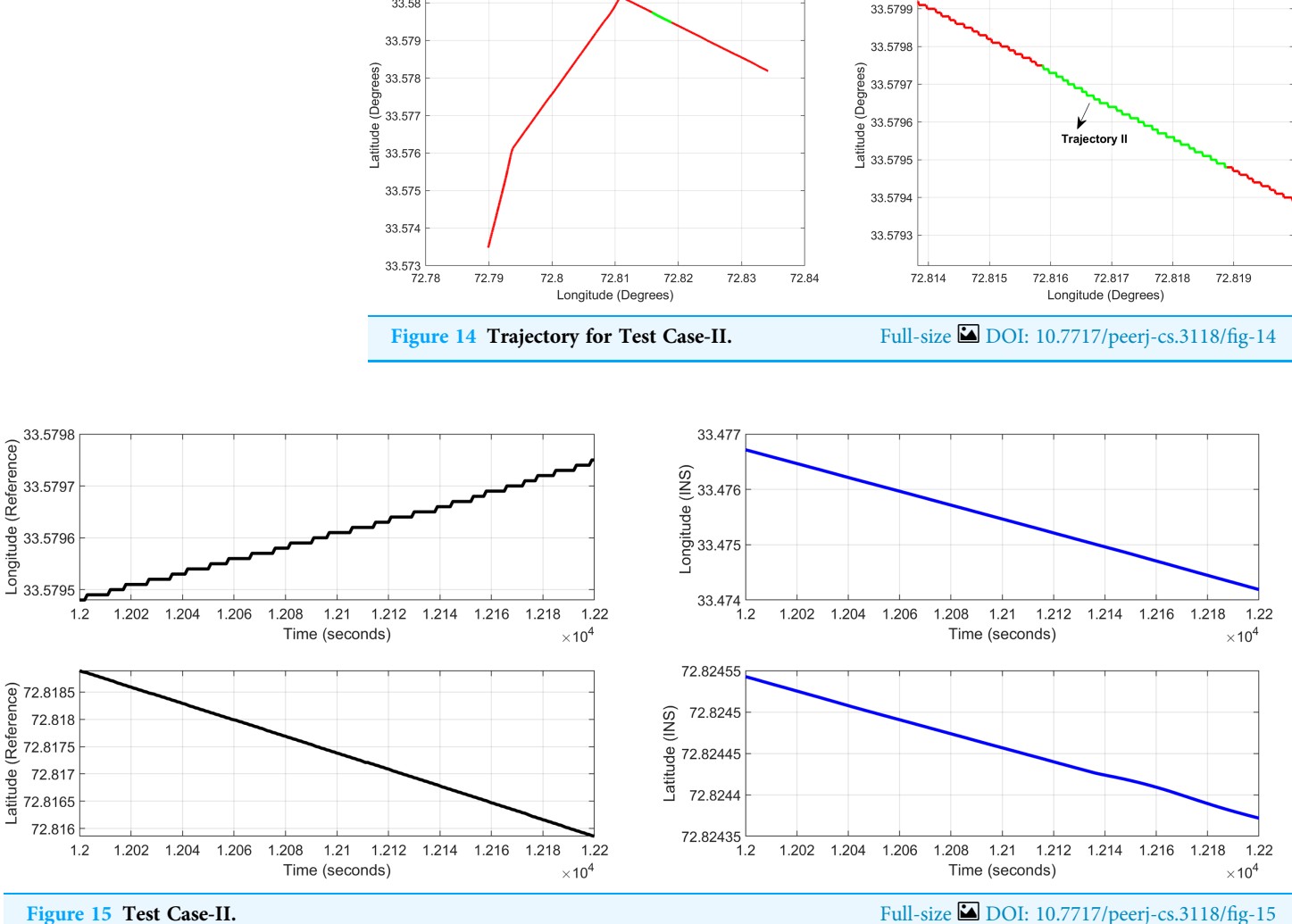

**Figure 14 Trajectory for Test Case-II.**

**Figure 15 Test Case-II.**

The values of the terrain roughness parameters for the two test regions are shown in Table 8. Where it can be observed that Test Region-II has more features compared to Test Region-I. The values of the metrics MAE, MSE and RMSE computed for two test cases are shown in Table 9 (with conventional MAD) and Table 10 (with fuzzy heuristic MAD). From the Table 9, it can be observed that the error values with the conventional MAD are lower for Test-Case-I compared to the position errors for Test-Case-II despite higher Sigma-Z and Sigma-T values. The reason is as follows: although, in general, the more unique terrain features, the higher is the positioning accuracy. However, the accuracy also depends on other factors such DEM-size and profile length selection (*Yousuf & Kadri, 2024*). In this article, for MAD correlation, the profile length 'l' is selected to be l = 10 for both test cases. The greater the profile length, the greater is the accuracy. Also, we used different DEM sizes for the two test cases. Based on the DEM size, no exact idea can be made as the accuracy may increase or decrease with minimizing or maximizing the DEM

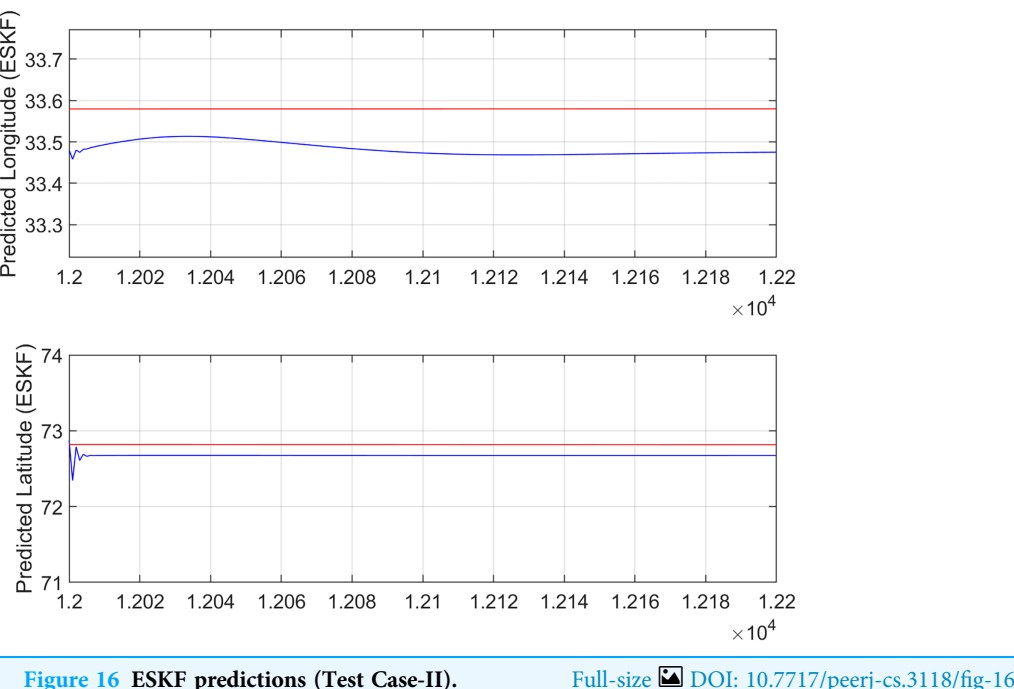

**Figure 16 ESKF predictions (Test Case-II).**

size. In Table 10, the error metrics for the proposed hybrid ESKF and FH-MAD are given. It can be observed that proposed method performs well in terms of position accuracy for the two DEMs having different terrain features with the errors being bounded within an acceptable range. However, the accuracy for Test Region-II is less compared to Test Region-I for the reason described above. Another reason could be the selection of the terrain related parameter 'L'. This parameter is depending on terrain characteristics. However, it is assumed to be a constant parameter in this article and has been selected empirically. The computational time with the proposed MAD algorithm and the conventional MAD algorithm for the two test cases for Region-I and Region-II is compared in Fig. 19.

## Comparison between conventional MAD and the proposed FH-MAD algorithm

The mean absolute difference (MAD) algorithm is frequently used for terrain matching in the standard terrain contour matching (TERCOM) system. The traditional MAD algorithm calculates the absolute difference between the corresponding elevation values at each position by sliding a reference terrain map over the elevation data obtained by the on-board sensor (usually from an altimeter or radar). The location that it produces the lowest MAD value is regarded as the most likely current position of the vehicle. The mean of these absolute differences is computed for every possible position. Because it must assess every potential match within a particular search area, regardless of terrain features or matching suitability, this thorough comparison process is computationally demanding. Particularly in real-time or resource-constrained applications, the deterministic approach
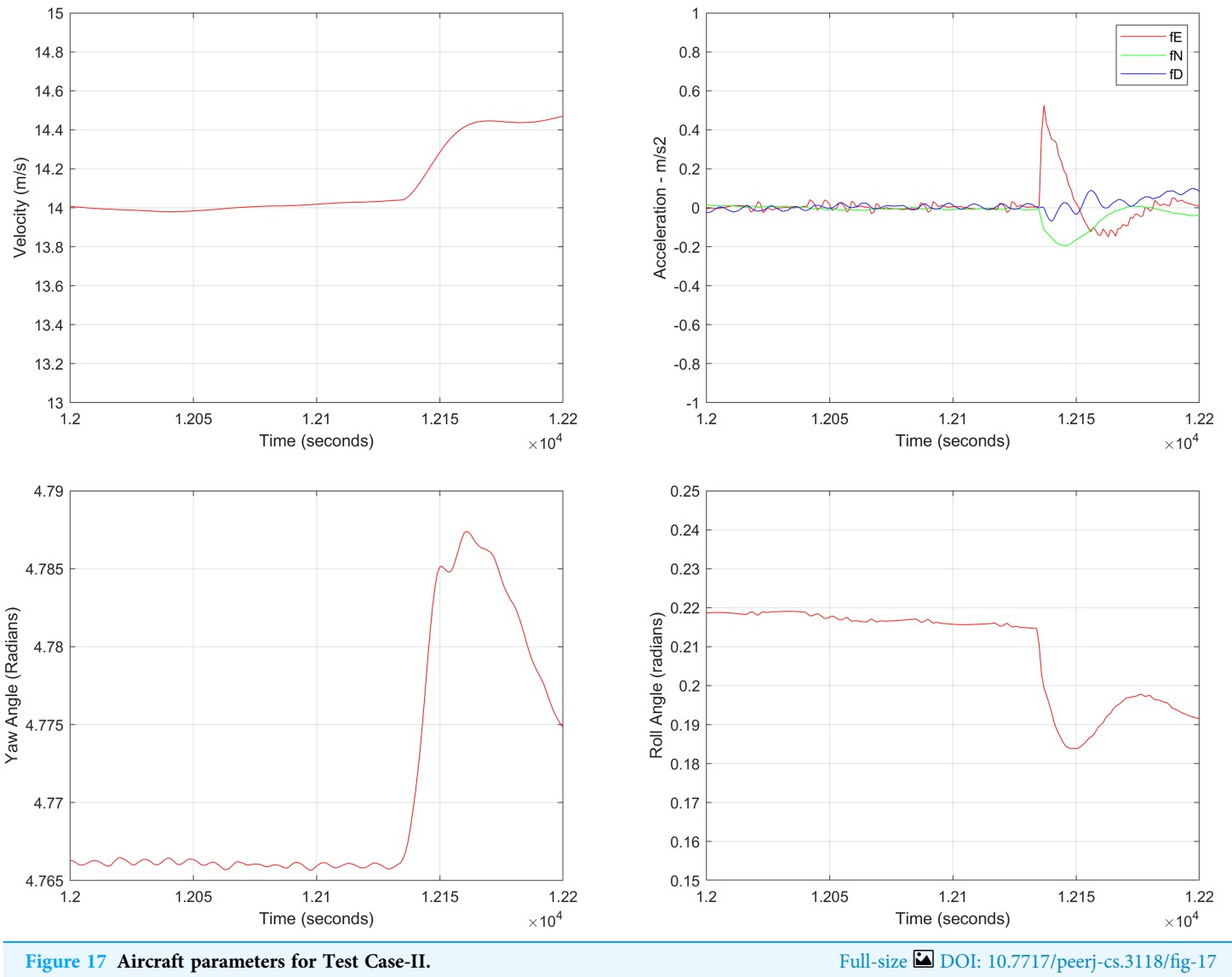

**Figure 17 Aircraft parameters for Test Case-II.**

may result in longer processing times and lower efficiency due to its inability to adjust to dynamic or ambiguous terrain.

The DEM size used in Test Case-II is greater than the DEM size in Test-Case-I. The conventional MAD therefore requires more time searching the DEM for Test-Case II compared to Test-Case-I. Similar trend can be observed with the proposed FH-MAD algorithm for the two regions. However, it can be clearly observed from the graph that the computational time is significantly reduced with the proposed FH-MAD correlation algorithm satisfying one of the major requirements for real-time performance. Furthermore, it must be highlighted that the quality of the predicted estimates from a TAN algorithm mainly depends on four factors: (1) the INS (2) the accuracy of the DEM (3) the terrain features and (4) the correctness of the altimeter measurements. The resolution of the DEM used in this work is 1 arc s. The predicted error for the latitude is up to five

 

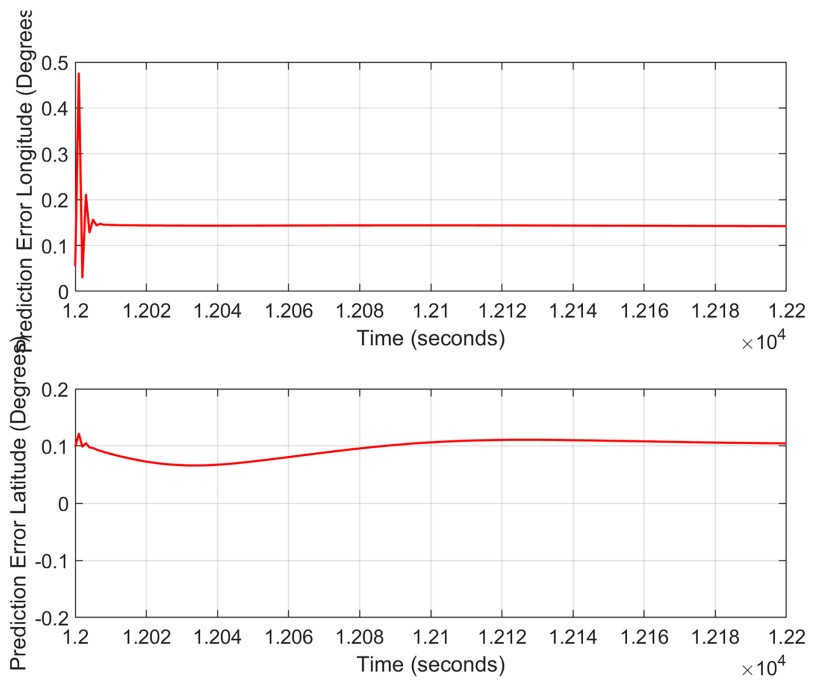

**Figure 18** ESKF prediction errors (Test Case-II).

**Table 8 Values for terrain roughness indicators.**

| Test case | Parameter | Values |
|---|---|---|
| I | Sigma-Z | 1,200.22 |
| | Sigma-T | 84.47 |
| II | Sigma-Z | 1,576.84 |
| | Sigma-T | 119.560 |

**Table 9 MAE, MSE, and RMSE errors for conventional MAD algorithm.**

| Test case | Method | Position | MAE (degrees) | MSE (degrees$^2$) | RMSE (degrees) |
|---|---|---|---|---|---|
| I | Conventional | Latitude | 0.00413 | 0.00017 | 0.00417 |
| | MAD | Longitude | 0.03470 | 0.00120 | 0.03470 |
| II | Conventional | Latitude | 0.01224 | 0.00018 | 0.01376 |
| | MAD | Longitude | 0.09565 | 0.022048 | 0.46956 |

**Table 10 MAE, MSE, and RMSE errors for optimized MAD algorithm (yaw and roll angle).**

| Test case | Method | Position | MAE (degrees) | MSE (degrees$^2$) | RMSE (degrees) |
|---|---|---|---|---|---|
| I | Optimized | Latitude | 0.00405 | 4.0333e−04 | 0.002008 |
| | MAD | Longitude | 0.00406 | 0.004089 | 1.67275e−5 |
| II | Optimized | Latitude | 0.14435 | 0.021512 | 0.146670 |
| | MAD | Longitude | 0.09608 | 0.009474 | 0.097335 |

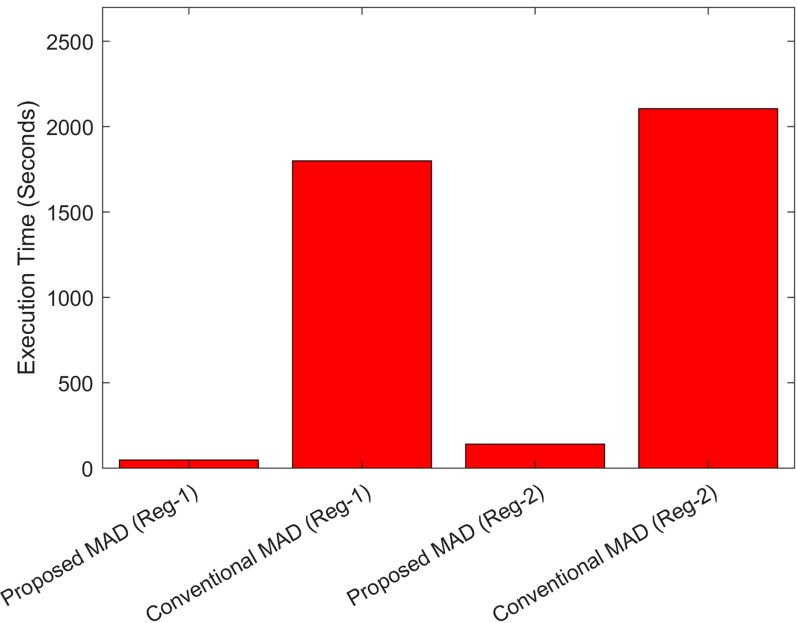

**Figure 19 Comparison of the execution times between conventional and proposed fuzzy heuristic MAD algorithm.**

decimal places (*i.e.*, 1 m). Also, the predicted error for the longitude is also up to five decimal places (*i.e.*, 1 m). The major reason for this error, despite the convergence trend of the proposed ESKF algorithm, is the non-availability of a high-quality DEM. Therefore, the future work is focused towards the task of incorporating a high-quality DEM with the proposed hybrid technique in order to test the levels of accuracies of the predicted target region.

## DISCUSSION

The proposed scheme in this work uses the error state Kalman filter. The ESKF proposed is selected due to the following advantages over the Indirect fuzzy robust cubature Kalman filter with normalized input parameters: Compared to the IFR-CKF, the error state Kalman filter (ESKF) has a number of benefits that make it more appropriate for real-time embedded applications and UAVs. Its linearized, simplified formulation results in lower-dimensional state updates and increased computational efficiency (*Yousuf & Kadri, 2024*). ESKF ensures stable performance by concentrating on small error estimates around a nominal state, making it ideal for high-rate systems. Additionally, it offers a clearer model structure that makes tuning easier, and improved numerical stability in nonlinear settings (*Tabassum et al., 2023*). ESKF reduces complexity and overfitting by not depending on fuzzy logic or heuristic tuning like IFR-CKF does (*Mascher et al., 2022*). In addition, it is technique that has shown dependability in robotics, aerospace, and GPS-INS integration scenarios.

Compared to LSTM, particle filter, and the conventional extended Kalman filter (EKF), the error state Kalman filter (ESKF) has a number of advantages. ESKF is model-based and

provides real-time, interpretable state estimation with low computational requirements, in contrast to LSTM, which is a model that requires intensive training. ESKF is much more scalable and efficient than particle filters, which makes it appropriate for embedded systems (*Yousuf & Kadri, 2024*). Without the complexity of particle resampling, it is simpler to implement and produces deterministic results. ESKF is more effective to EKF because it handles non-linearities more accurately and has better numerical stability, especially when it comes to orientation estimation (*Tabassum et al., 2023*). ESKF achieves better performance and modularity by linearizing the error state instead of the full state, which makes it perfect for complex, real-world scenarios like robotics or UAV navigation (*Mascher et al., 2022*).

## CONCLUSIONS

The main purpose of this work was to design and demonstrate a Terrain Reference Navigation (TRN) scheme for determining the aircraft position using a low-cost Digital Elevation Map (DEM). In this work, a fuzzy logic based correlation scheme is proposed for the selection of optimal matching area utilizing the aircraft heading and roll angle parameters. It was found that the fuzzy logic approach makes the search process more reasonable as compared to the conventional MAD approach with significant reduction in the computational time. Furthermore, the TERCOM algorithm was developed using the error state Kalman filter for aircraft localization for the optimal estimation of state errors. A low-cost strap-down INS for the system was implemented. The derived expressions were investigated under different circumstances including different aircraft velocities and heading angles. From the simulation tests, it was found that the presented scheme produces effective results under varying conditions of aircraft maneuver. The algorithm is tested under two different cases with different regions having variations in topographical features and different trajectories, time durations and different heading and velocity ranges. In addition to the tuning of the conventional filter metrics in the ESKF such as error covariance P, process noise covariance Q and measurement noise covariance R, the choice of two other constants are also studied in this scheme, *i.e.*, the gain coefficient 'k' and the filter shaping coefficient '$\beta$' in addition to the process, measurement and error noise covariance matrices in the filter design. The optimal values of the coefficients are selected after a number of iterations.

The prediction errors are quantified in terms of three most widely used metrics in literature, *i.e.*, MSE, MAE and RMSE. The simulation results show that the ESKF strategy with the proposed fuzzy logic based correlation approach not only provides good tracking accuracy but also significantly reduces the computational cost compared to the conventional TERCOM algorithm. One of the major drawbacks of the proposed method is that the terrain parameter used for constructing the membership functions (MFs) for the test cases is considered to be constant. In future work, a mathematical formulation for the computation of terrain parameter will be developed studying the effectiveness of the single as well as the fusion of terrain characteristics such as DEM slope, aspect and profile curvature *etc*. The future work will also be focused on the development of a combined UKF and particle filter based algorithm replacing the ESKF in the current scheme.

### Funding

Prince Sultan University, Riyadh, Saudi Arabia funded the Article Processing Charges (APC) of this publication. The funders had no role in study design, data collection and analysis, decision to publish, or preparation of the manuscript.

### Grant Disclosures

The following grant information was disclosed by the authors:
Prince Sultan University, Riyadh, Saudi Arabia, Article Processing Charges (APC).

### Competing Interests

The authors declare that they have no competing interests.

### Author Contributions

- Muhammad Bilal Kadri conceived and designed the experiments, performed the experiments, analyzed the data, performed the computation work, authored or reviewed drafts of the article, and approved the final draft.
- Sofia Yousuf conceived and designed the experiments, performed the experiments, analyzed the data, performed the computation work, prepared figures and/or tables, and approved the final draft.

### Data Availability

Code is available at Zenodo:

Muhammad Bilal Kadri. (2025). bilalkadri/Hardware-Implementation-TAN-EsKF: Initial release (v1.0.0). Zenodo. https://doi.org/10.5281/zenodo.15343700.

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
