# Peer review of "An advanced error state Kalman filter (ESKF)-based terrain contour matching (TERCOM) method for tracking an aerial vehicle using a low-cost digital elevation map"

_PeerJ Computer Science, doi:10.7717/peerj-cs.3118_

## Round 0.1 · original submission · Major Revisions

Both reviewers have provided helpful feedback.

Reviewer 1 ·

Basic reporting

-

Experimental design

-

Validity of the findings

-

Additional comments

1. The proposed method is for the multi-sensor integrated system. However, the specific nonlinear system state and measurement equations are not provided, including the state transition and measurement matrices.

2. The authors list 5 contributions. However, most of them are not fundamental and novel.

3. The linearization of the system must be conducted based on the specific system state and measurement equations.

4. Fig. 1 does not reflect/highlight the proposed fuzzy logic method.

5. What is the B matrix in (20)? Please provide the specific form.

6. As shown in (24), the measurement matrix is a unit matrix, which is incorrect. It should be derived via the linearization process of the specific nonlinear measurement function.

7. The proposed fuzzy logic seems to have nothing to the Kalman filter. Then, how to ensure the Kalman filtering accuracy, which is influenced by uncertainties such as system model error, measurement outliers, and unknown system noise statistics?

8. The fuzzy logic method in Section 3.5 is not elaborated. What are the fuzzy rules established?

9. Only simulation analysis is provided, while experimental results are missing. Comparison analysis of advanced techniques such as the Indirect fuzzy robust cubature Kalman filter with normalized input parameters.

10. Advanced Kalman filters, such as improved CKFs and UKFs, were developed in the literature (although they were developed for broad vehicle navigation, they can also be applied to TRN). However, they are are not introduced – e.g., A robust Kalman filter based on kernel density estimation for system state estimation against measurement outliers, Set-membership based hybrid Kalman filter for nonlinear state estimation under system uncertainty, Cubature rule- based distributed optimal fusion with identification and prediction of kinematic model errors for integrated UAV navigation, Adaptively random weighted cubature Kalman filter for nonlinear systems, Indirect fuzzy robust cubature Kalman filter with normalized input parameters, An advanced cubature information filtering for indoor multiple wideband source tracking with a distributed noise statistics estimator, Random weighting-based nonlinear Gaussian filtering, A novel fitting H-infinity Kalman filter for nonlinear uncertain discrete-time systems based on fitting transformation, Maximum likelihood principle and moving horizon estimation based adaptive unscented Kalman filter, Interacting multiple model estimation-based adaptive robust unscented Kalman filter, Modified strong tracking unscented Kalman filter for nonlinear state estimation with process model uncertainty, Sage windowing and random weighting adaptive filtering method for kinematic model error, Windowing-based random weighting fitting of systematic model errors for dynamic vehicle navigation. Please introduce advanced Kalman filters to strengthen the literature.

11. As mentioned previously, the proposed system is a multi-sensor integrated navigation system. However, advanced techniques on multi-sensor integrated navigation system are not introduced – e.g., Robust unscented Kalman filtering with measurement error detection for tightly coupled INS/GNSS integration in hypersonic vehicle navigation, Unscented Kalman filter with process noise covariance estimation for vehicular INS/GPS integration system, Covariance matching based adaptive unscented Kalman filter for direct filtering in INS/GNSS integration, A new direct filtering approach to INS/GNSS integration, A derivative UKF for tightly coupled INS/GPS integrated navigation, Model predictive based unscented Kalman filter for hypersonic vehicle navigation with INS/GNSS integration, Robust unscented Kalman filter based decentralized multi-sensor information fusion for INS/GNSS/CNS integration in hypersonic vehicle navigation, Cubature Kalman filter with closed-loop covariance feedback control for integrated INS/GNSS navigation, Random weighting estimation for fusion of multi- dimensional position data, Random weighting method for multi-sensor data fusion, Multi- sensor data fusion for INS/GPS/SAR integrated navigation system. Please introduce the advanced techniques to strengthen the literature survey.

Reviewer 2 ·

Basic reporting

A. General

This paper proposes a Terrain-Aided Navigation (TAN) system for UAVs by integrating a Fuzzy Heuristic Mean Absolute Deviation (FH-MAD) approach with an Error State Kalman Filter (ESKF)-based TERCOM algorithm. The proposed system is designed to enhance tracking accuracy and computational efficiency, and its evaluation in a simulated environment shows promising outcomes. With refinement in a few key areas, this work has the potential to make a meaningful contribution to terrain-based navigation research.

B. Major comments

1. While the derivation of the ESKF is well-structured, some symbols are either undefined or used inconsistently. It would be helpful to standardize the notation and provide clear definitions where needed to enhance clarity.

2. The current two-row formatting of the state vector in Equation (19) aids readability, but revising it to follow the conventional n×1 Kalman filter format would improve consistency with standard practices.

3. The use of fuzzy logic for region selection is creative. Providing further explanation on how the input/output membership functions were chosen and how they compare with alternative approaches could strengthen the methodology.

4. The manuscript references filters such as EKF, PF, and LSTM. Including a brief comparison, either through discussion or experimental benchmarks, could help position the ESKF performance within the broader context.

5. Figure 31 includes a comparison with a conventional method, but the details of this baseline are not clearly outlined. Providing a brief description would support a more intuitive and transparent comparison.

6. The term “low-cost” in the title and text could benefit from clarification. Briefly describing the source, resolution, or cost characteristics of the DEM would reinforce this claim.

7. Improving the flow between sections, particularly when moving between fuzzy logic, DEM handling, and filtering, could make the paper more cohesive for readers.

C. Minor comments and corrections
1. Strengthening the alignment between the process blocks in Figure 1 and the accompanying equations would help readers better understand the system architecture.

2. Please correct the typo “serach” to “search” in the abstract (PDF page 5).

3. Table 6 refers to matrix R as 15×15, while later text indicates a 3×3 dimension. Ensuring consistency in such technical details will improve clarity.

4. Enhancing figure resolution and ensuring axis labels, units, and legends are consistently included would improve readability.

5. A few symbols and terms differ from commonly accepted conventions. A brief clarification or alignment with standard notation would be appreciated.

6. A thorough proofreading to check for spacing after citations and consistent punctuation would further polish the manuscript

Experimental design

.

Validity of the findings

.

Additional comments

.

---

## Round 0.2 · Minor Revisions

Dear authors,

You are advised to critically respond to the reviewer's comments point by point when preparing a new version of the manuscript and while preparing for the rebuttal letter.

Please address all comments/suggestions provided by reviewers, considering that these should be added to the new version of the manuscript.

Kind regards,
PCoelho

Reviewer 1 ·

Basic reporting

The paper has been significantly improved, and my concerns have been carefully addressed as well. I would recommend that the paper be accepted for publication.

Experimental design

-

Validity of the findings

-

Reviewer 2 ·

Basic reporting

This paper proposes an ESKF-based Terrain Aided Navigation (TAN) system incorporating a Fuzzy Heuristic Mean Absolute Deviation (FH-MAD) algorithm, designed to enhance UAV localization accuracy and reduce computation time using a low-cost Digital Elevation Map (DEM). The integration of fuzzy logic for terrain correlation and ESKF for state estimation is novel and shows promising results in simulation.

Experimental design

-

Validity of the findings

-

Additional comments

1) In Equation (27), the error-state vector is defined as having two primary components: (1) position, velocity, attitude (PVA), and (2) sensor biases (accelerometer and gyroscope). However, the state vector shown in Equation (27) differs from the one used in the state transition matrix in Equation (29), as well as the vector shown later in Equation (31). To maintain consistency, the state vector should explicitly follow the structure or a comparable clear format aligned across all relevant equations:

x=〖[p_3x1 v_3x1 a_3x1 〖f_b〗_3x1 〖w_b〗_3x1]〗^T

2) Related to the first comment, the rationale behind dividing the state vector into two elements (pva and biases) should be elaborated. Is this decomposition for modeling simplicity, numerical efficiency, or observability reasons?

3) In Equation (29), the state transition matrix

---

## Round 0.3 · accepted · Accept

Dear authors, we are pleased to verify that you have met the reviewers' valuable feedback to improve your research.

Thank you for considering PeerJ Computer Science and submitting your work.

Kind regards
PCoelho